



# Nutrients attenuate the negative effect of ocean acidification on reef coral calcification in the Arabian Sea upwelling zone (Masirah Island, Oman)

Philipp M. Spreter[1], Markus Reuter[1], Regina Mertz-Kraus[2], Oliver Taylor[3], Thomas C. Brachert[1]

[1]Institut für Geophysik und Geologie, Universität Leipzig, Talstraße 35, 04103 Leipzig, Germany
[2]Institut für Geowissenschaften, Johannes Gutenberg-Universität Mainz, Johann-Joachim-Becher-Weg 21, 55128 Mainz, Germany
[3]Five Oceans Environmental Services, Villa 1756, Way 3021; Shatti Al Qurm, Muscat, Sultanate of Oman

*Correspondence to*: Philipp M. Spreter (philipp.spreter@uni-leipzig.de)

**Abstract.** Tropical shallow-water reefs are the most diverse ecosystem in the ocean. Its persistence rests upon adequate calcification rates of the reef building biota, such as reef corals. Optimum calcification rates of reef corals occur in oligotrophic environments with high seawater saturation states of aragonite ($\Omega_{sw}$), which leads to increased vulnerability to anthropogenic ocean acidification and eutrophication. The calcification response of reef corals to this changing environment is largely unknown, however. Here, we present annually and sub-annually resolved records of calcification rates (n=3) of the coral *Porites* from the nutrient rich and low $\Omega_{sw}$ Arabian Sea upwelling zone (Masirah Island, Oman). Calcification rates were determined from the product of skeletal extension and bulk density derived from X-ray densitometry. Compared to a reference data set of coral skeletons from typical reef environments (Great Barrier Reef, Hawaii), mean annual skeletal bulk density of *Porites* from Masirah Island is reduced by 28 %. This density deficit prevails over the entire year and probably reflects a year-round low saturation state of aragonite at the site of calcification ($\Omega_{cf}$), independent of seasonal variations in $\Omega_{sw}$ (e.g. upwelling). Mean annual extension rate is 20 % higher than for the reference data set. In particular, extension rate is strongly enhanced during the seasons with the lowest water temperatures, presumably due to a high $PO_4^{3-}/NO_3^{-}$-ratio promoting rapid upward growth of the skeleton. Enhanced annual extension attenuates the negative effect of low density on calcification rate from -25 % to -11 %, while sub-annual calcification rates during the cool seasons even exceed those of the reference corals. We anticipate optimal nutrient environments (e.g. high $PO_4^{3-}/NO_3^{-}$-ratios) to have significant potential to compensate the negative effect of ocean acidification on reef coral calcification, thereby allowing to maintain adequate rates of carbonate accumulation, which are essential for preserving this unique ecosystem.

## 1 Introduction

The calcareous skeletons of zooxanthellate scleractinian corals (reef corals) are the basic building blocks of tropical shallow-water reefs, the most diverse ecosystem in the ocean (Hughes et al., 2017). The persistence of reef habitats rests upon





coralline aragonite precipitation (calcification), which determines the delicate balance between reef build-up and natural destruction. Optimum calcification of reef corals is found in oligotrophic water masses of temperatures between 21 °C to 29.5 °C and a saturation state of seawater with respect to aragonite >3.3 ($\Omega_{sw}$) (Kleypas et al., 1999). Anthropogenic greenhouse gas emissions threaten reef coral calcification by increasing sea surface temperature (SST) and by causing ocean

acidification. In addition, land use, sewage disposal and fish farming turn the near-shore shallow-marine environments towards more eutrophic conditions. The response of reef coral calcification on this rapidly changing environment remains contemporarily a matter of debate.

Coral calcification rate (g cm$^{-2}$ yr$^{-1}$) is the product of linear extension rate (cm yr$^{-1}$) measured along the axis of maximum growth and bulk density (g cm$^{-3}$) of an annual skeletal growth increment (Dodge and Brass, 1984). Bulk density within an

annual increment is spatially not uniform, and portions of elevated density (high-density bands = HDBs) reflect biomineralization at less optimal high and low growth temperatures (Highsmith, 1979; Klein and Loya, 1991). While on an annual time-scale, extension rate and calcification rate is positively related to SST, skeletal bulk density is negatively related to SST (Fabrizius et al., 2011; Lough, 2008; Lough and Barnes, 2000). These inverse linear relations of extension rate and skeletal density at increasing temperature turn over at a certain, taxon-specific, threshold temperature, when extension rates

rapidly decline due to increasing thermal stress (Cantin et al., 2010). In addition to temperature, the aragonite saturation of the calcifying fluid ($\Omega_{cf}$) determines bulk skeletal density (Mollica et al., 2018). $\Omega_{cf}$ is approximately five times higher than the aragonite saturation of the external seawater ($\Omega_{sw}$) and long-term changes in $\Omega_{sw}$ due to ocean acidification lead to declining $\Omega_{cf}$ (McCulloch et al., 2017; D'Olivo et al., 2019). On an intra-annual basis, however, corals are able to maintain relatively stable levels of $\Omega_{cf}$ largely independent of short term variations in $\Omega_{sw}$ by upregulating their internal $pH_{cf}$ and

$DIC_{cf}$ pool (McCulloch et al., 2017; DeCarlo et al., 2018; D'Olivo and McCulloch, 2017, Ross et al., 2019a). With regard to climate models for the end of the 21st century, McNeil et al. (2004) stated a general increase in calcification rate as the effect of ocean warming far outweighs deficits due to decreases in seawater aragonite saturation.

Near-shore coral reefs are increasingly exposed to anthropogenic eutrophication induced by land use, fish farming and savage disposal (Lapointe and Clark, 1992; Chen et al., 2019; Chen and Yu, 2011). The effects of nutrients on coral reefs

appear to be versatile, however. Reef corals are highly adapted to oligotrophic waters because symbiosis with phototrophic zooxanthellae allows an efficient use of essential nutrients and to outcompete other fast-growing biota on a reef whose growth is inhibited by the undersupply of nutrients (Vermeij et al., 2010; Barrot and Rohwer, 2012). Strong eutrophication disturbs this adaptive advantage, leading to harmful algal blooms followed by reef coral mass mortality (Al Shehhi et al., 2014) and reef destruction due to the increasing abundance of bioeroders (Hallock, 1988). On the other hand, moderate

increases of certain nutrients such as ortho-phosphate ($PO_4^{3-}$) have been shown to promote linear extension rates but to inhibit skeletal thickening, thus having a negative effect on skeletal density (Koop et al., 2001; Dunn et al., 2012; Bucher and Harrison, 2001). The opposite effect is reported for nitrate ($NO_3^-$), even though the calcification response is less pronounced compared to $PO_4^{3-}$ (Koop et al., 2001). Increasing eutrophy is considered to explain enhanced coral calcification rates along



an offshore – inshore gradient where the effect of temperature is negligible (D'Olivo et al., 2013; Risk and Sammarco, 1991;
Manzello et al., 2015).

The versatile effect of nutrients in interaction with the impeding effect of low aragonite saturation on coral calcification remains largely unknown. Study areas well suited for this kind of research are those affected by upwelling, since upwelling deep water masses are rich in nutrients but cool and low in $\Omega_{sw}$. Upwelling areas are heralded as possible refuges for tropical coral species in times of globally rising temperatures due to the potential to mitigate high SSTs (Riegl & Piller, 2002;
Chollett et al., 2010). Furthermore, the calcification response of reef corals from upwelling zones could provide important implications regarding the survival of tropical coral reefs, which are increasingly threatened by human caused ocean acidification and eutrophication. Published calcification data of a major reef-building coral genus from the Indo-Pacific (*Porites*) growing within regions affected by upwelling are sparse. Only two records of extension rate and skeletal density are available from the eastern Pacific upwelling zones (Manzello et al., 2015; Mollica et al., 2018). Solely extension rates are
available from the upwelling regions of the northwest Indian Ocean, which do not allow for the determination of reef coral calcification rates (Montone, 2010; Tudhope et al., 1996; Watanabe et al., 2017). In the here presented study, we report the first annually and monthly resolved calcification records (extension rate, skeletal bulk density, calcification rate) of three *Porites* coral specimens from the Arabian Sea upwelling zone (Masirah Island, Oman). The results are discussed with regard to environmental controls on calcification (SST, $\Omega_{sw}$, nutrients) and compared to coral calcification data reported in the
literature from typical reef environments (warm, oligotrophic). This study contributes to assessing changes in patterns of reef coral calcification in the context of ongoing global change (ocean acidification and eutrophication) and contributes to the identification of regions deserving high conservation priority due to favourable environmental conditions for the persistence of tropical shallow-water reefs.

## 1.1 Arabian Sea climate and oceanography

The northwestern portion of the Indian Ocean between India and the Arabian Peninsula is the Arabian Sea. Its coastline against the Arabian Peninsula forms a straight line running towards the northeast up to the Cape Ras-al-Hadd where it turns almost at right angles to the northwest into the Gulf of Oman (Fig. 1). The regional climate of the Arabian Sea is characterised by a semi-annual alternation of the prevailing wind directions (Beal et al., 2013). During northern hemisphere summer, strong winds of the Southwest Monsoon (SWM) cross the Arabian Sea in the direction of the low pressure system
above the Tibetan Plateau (Findlater, 1969). During winter, prevailing winds from the northeast cause the Northeast Monsoon (NEM) (Hastenrath and Greischar, 1991). Low wind speeds without preferred orientation typically occur during the two intermonsoon seasons (spring intermonsoon = SIM; autumn intermonsoon = AIM) (Beal et al., 2013; Lee et al., 2000). Surface wind fields are the driving force behind the upper hydrospheric structure and the seasonal variation of the oceanic surface current system (Swallow and Bruce, 1966). During summer, southwest monsoonal winds cause a strong
coastal current (Oman Coastal Current) which runs northward parallel to the coast of the Arabian Peninsula and induces rigorous upwelling (Currie et al., 1973; Currie, 1992, Smith and Bottero, 1977). Increased nutrient supply during southwest



monsoonal upwelling is associated with an increased primary productivity in the euphotic zone (Anderson et al., 1992; Bauer et al., 1991) (Fig. 2). Compared to equatorial upwelling regions of the eastern Pacific, the northern Arabian Sea upwelling is characterized by a high phosphate to nitrate ratio (Kleypas et al., 1999). But although nutrient supply is linked to upwelling,

concentrations of $PO_4^{3-}$ and $NO_3^-$ remain on a high level throughout the year, because of a prevailing iron limitation of the primary production (Mother Earths Iron Experiment) (Smith, 2001). Southwest monsoonal upwelling furthermore causes a drop in surface water pH, causing seawater aragonite saturation ($\Omega_{sw}$) to decrease temporally from 3.5 – 4 during non-upwelling season to 2.5 during the upwelling season, which is well below critical values assumed to be required for coral growth (Kleypas et al., 1999; Omer, 2010).

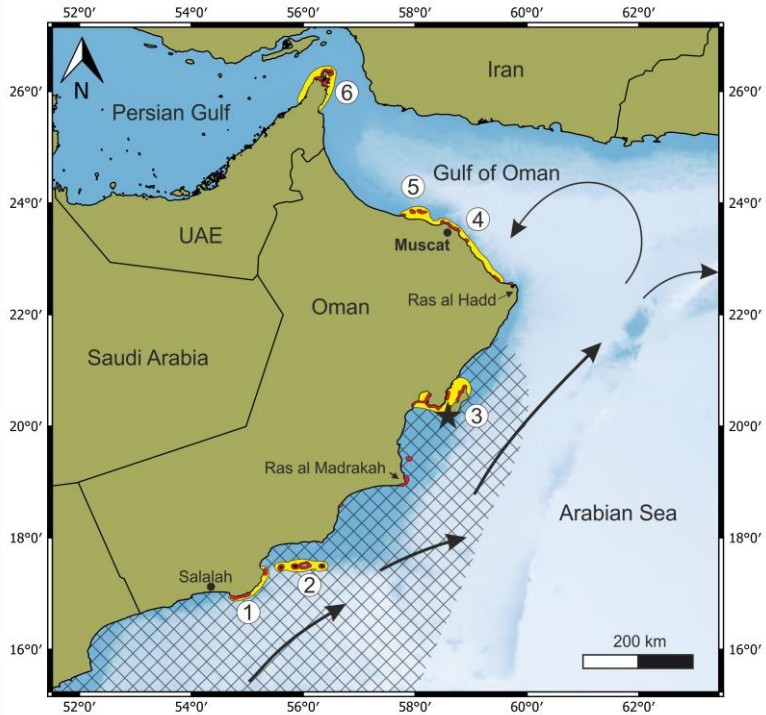


**Figure 1: General map of the eastern Arabian Peninsula and the adjacent seas showing the reef sites (red) and coral reef provinces off Oman (yellow) [(1) Marbat, (2) Kuria Muria Islands, (3) Masirah Island and Barr al Hikman, (4) Capital Area (5) Daymaniyat Islands and (6) Musandam (Salm, 1993; Burt et al., 2016)]. The black asterisk marks the sampling site at the southern tip of Masirah Island. Black arrows indicate the Omani current of the southwest monsoon season. The hatched area delimitates the**
**region of maximum upwelling inferred from August sea surface temperatures <25 °C (1995 – 2005 monthly averages; WOA18, Locarnini et al., 2018). The blue shading displays water depth gradients (dark blue = shallow, light blue = deep).**



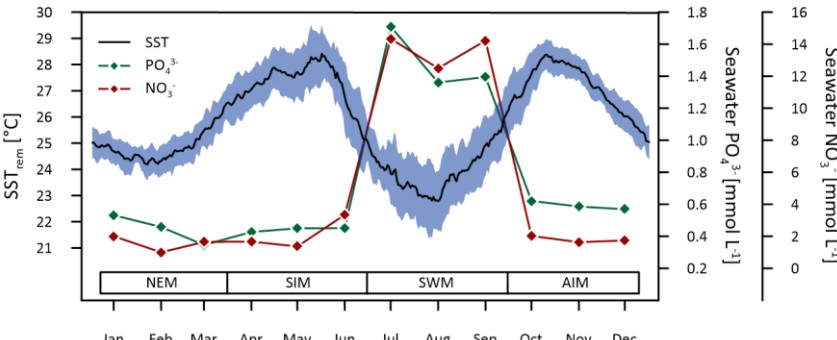

**Figure 2: Ocean climatological data estimated from daily SSTs (JPL MUR) and monthly seawater phosphate (PO₄³⁻) and nitrate (NO₃⁻) concentrations (WOA18) at the sampling site. The blue shaded area represents the deviation (1σ) from averages of daily SSTs within the period 2003 – 2018.**

## 1.2 Coral growth at Masirah Island

Coral growth off Oman occurs within six demarcated provinces, including Marbat, the Kuria Muria Islands, Masirah Island and Barr al Hikman, the Capital Area, the Daymaniyat Islands and Musandam (Fig. 1; Burt et al., 2016). Sheltered from the open ocean, coral growth at Masirah Island occurs to the west (Strait of Masirah) and southwest (Barr al Hikman) of the island (Salm, 1993). Compared to the other coral reef provinces of Oman, the reefs to the southwest of Masirah Island represent the largest area of spatially continuous coral cover, but have the lowest coral diversity (Salm, 1993; Coles, 1996). Fast growing cabbage and brain corals such as *Montipora* and *Platygyra* are the dominant genera, but massive *Porites* are also present (Coles, 1996).

## 2 Material and methods

In March 2018, three massive Porites-colonies (Sample identifier: 5.10; 5.13; 5.21) were collected at the southernmost tip of Masirah Island (20.16N, 58.64E) (Fig. 1). Because the sampling position on the uppermost part of the shore is not affected by common high tides, the corals must have been washed ashore during a high-energy storm or tsunami event. The last significant storm event was cyclone Gonu in 2007 with maximum wind speeds of 260 km/h. However, no immediate devastation caused by this storm was reported from Masirah Island, since most intense landfalls of the storm occurred at Ras al Hadd (Fig. 1) and further to the northwest in the Gulf of Oman (Fritz et al., 2010). In contrast, the second strongest cyclonic event reported for the Arabian Peninsula occurred in 1977, causing severe damage at Masirah Island by south-westerly winds with speeds of up to 185 km/h (Membery, 2002).

Coral samples were cut to slices of 6 mm thickness parallel to the axis of maximum growth using a rock saw at lowest tournament speed and equipped with a water-cooled diamond blade. Subsequent usage of a CNC mill ensured co-planarity with maximum deviations of 1 – 3 % over the entire slab. The slabs were ultrasonically cleaned in deionized water and dried overnight at 40 °C. Coral slabs were X-rayed using a digital X-ray cabinet (SHR 50 V) to document alternating growth



bands of high (HDB) and low density (LDB) (Knutson et al., 1972), biogenic borings, encrustations, and cementation.
Sampling transects for all further analyses were carefully selected so as not to be affected by bioerosion and encrustations
but normal to HDBs and LDBs following trajectories of maximum linear extension (for the positioning of individual
sampling transects see supplementary material, Fig. S1). Density measurements were performed using X—ray densitometry
based on CoralXDS software (Helmle et al., 2002, 2011). Grey-scale – density calibrations were verified by measurements
of standards for zero density (air, $\rho = 0$ g/cm$^{-3}$) and massive aragonite (Tridacna shell, $\rho = 2.93$ g/cm$^{-3}$) having the same
thickness as the coral slabs. Maximum target deviations were $0.02 \pm 0.01$ g cm$^{-3}$ for zero density (air) and $0.03 \pm 0.06$ g cm$^{-3}$
for massive aragonite (Tridacna shell). Width of density measurement transects were set to 4 mm, including a representative
mixture of approximately 12 corallites (4 x 6 mm).

Li/Mg thermometry was used for estimating absolute growth temperatures (Montagna et al., 2014; Cuny-Guirriec et al.,
2019, Ross et al., 2019b, Harthone et al., 2013; Fowell et al., 2016; D'Olivo et al., 2018; Zinke et al., 2019). Li/Mg – SST
relationships were shown to be site dependent, however, and similar Li/Mg-ratios produce differences in SST estimations of
~2 °C between inter-reef (Hathorne et al., 2013) and intra-reef settings (Fowell et al., 2016). Such spatial variability in Li/Mg
– SST relationships is likely due to poorly constrained effects of extension rate and seawater pH on skeletal Li/Mg ratios
(Fowell et al., 2016; Inoue et al., 2007; Tanaka et al., 2015). For this reason, we use a separate calibration of the Li/Mg
thermometer for Masirah Island corals in order to overcome misleading SST estimates resulting from local seawater pH and
extension rate effects associated with upwelling.

Trace and minor element concentrations were determined at the Institute for Geosciences, Johannes Gutenberg University
Mainz (Germany), using an Agilent 7500ce inductively coupled plasma-mass spectrometer (ICP-MS) coupled to an ESI
NWR193 ArF excimer laser ablation (LA) system equipped with a TwoVol2 ablation cell. The ArF LA system was operated
at a pulse repetition rate of 10 Hz and an energy density of c. 3 J cm$^{-2}$. Ablation was carried out under a He atmosphere and
the sample gas was mixed with Ar before entering the plasma. Measurement spots with a beam diameter of 120 µm were
aligned along transects in spot mode with a midpoint distance of 250 µm following discrete skeletal elements. Backgrounds
were measured for 15 s prior to each ablation. Ablation time was 30 s, followed by 20 s of wash out. The isotopes monitored
were 7Li, 25Mg, 43Ca and 138Ba. Signals were monitored in time-resolved mode and processed using an in-house Excel
spreadsheet (Jochum et al., 2007). Details of the calculations are given in Mischel et al. (2017). NIST SRM 610 and 612
were used as calibration material, applying the reference values reported in the GeoReM database (http://georem.mpch-
mainz.gwdg.de/, Application Version 27; Jochum et al., 2005; Jochum et al., 2011) to calculate the element concentrations
of the sample measurements. During each run, basaltic USGS BCR-2G, synthetic carbonate USGS MACS-3 and a nano-
powder pellet of biogenic carbonate JCp-1 were analyzed repeatedly as quality control materials (QCM) to monitor precision
and accuracy of the measurements as well as calibration strategy. All reference materials were analyzed at the beginning and
at the end of a sequence and after ca. 40 spots on the samples. For all materials 43Ca was used as internal standard applying
for the USGS BCR-2G and MACS-3 the preferred values reported in the GeoReM database, for JCp-1 38.18 wt.% (Okai et
al., 2002) and for the samples a Ca content of 39 wt.% (Mertz-Kraus et al., 2009). Resulting element concentrations for the





QCMs together with reference values are provided in the supplementary material (Table S1). Element concentrations for the samples are converted into molar ratios of Ca, i.e., Li/Ca, Mg/Ca, Ba/Ca as well as Li/Mg.

Daily Sea surface temperatures ($SST_{rem}$) were extracted from JPL MUR (v4.1) available from https://podaac.jpl.nasa.gov. The JPL MUR data range used for this study covers the period 2003 – 2018 and has a spatial resolution of 0.01° degrees (Grid cell: N19.9, E58.6). SSTs of equal calendar dates of consecutive years were averaged receiving one generalized annual record of mean daily SSTs for the period 2003 – 2018 (Fig. 2). Reliability of the remote sensed data was confirmed by daily in-situ observed SSTs ($SST_{in-situ}$) recorded at the southern tip of Masirah Island (water depth: 5 m) between October 2001

and September 2002 (Wilson, 2007). Annual mean SSTs were in excellent accordance to each other ($SST_{rem}$ = 25.79 °C ; $SST_{in-situ}$ = 25.54 °C) and daily SSTs were strongly correlated ($r^2$ = 0.79 , p < 0.0001).

Seawater phosphate and nitrate concentrations were extracted from the World Ocean Atlas 2018 (WOA18) available on https://www.nodc.noaa.gov/OC5/woa18/woa18data.html. The WOA18 nutrient data is a generalized interpolation of all available in-situ observations performed within individual months at certain depth levels within each 1° square (Garcia et al.,

2018). Annual mean nutrient concentrations were extracted for Masirah Island as well as for upwelling sites from where published coral calcification data was available (Table 1). Monthly values were solely extracted for Masirah Island (Grid cell: N19.5, E58.5, water depth: 5 m; Fig. 2).

**Table 1: Mean annual seawater phosphate ($PO_4^{3-}$) and nitrate ($NO_3^-$) concentrations (±1σ) of Masirah Island as well as of upwelling sites from where published coral calcification data is available (WOA18). Numbers (n) indicate the quantity of observations used for the calculation of an average concentration for each grid square.**

| Location | Grid cell | Water depth [m] | $PO_4^{3-}$ [μmol L$^{-1}$] | n | $NO_3^-$ [μmol L$^{-1}$] | n | $PO_4^{3-}$ / $NO_3^-$ |
|---|---|---|---|---|---|---|---|
| **Masirah Island** (this study) | 58.5E ; 19.5N | 5 | 0.741 ± 0.616 | 11 | 4.790 ± 6.257 | 2 | 0.155 |
| **Marbat** (Tudhope et al., 1996) | 54.5E ; 16.5N | 5 | 0.478 ± 0.302 | 6 | 2.513 ± 1.558 | 3 | 0.190 |
| **Kuria Muria** (Montone, 2010) | 56.5E ; 19.5N | 5 | 0.472 ± 0.074 | 9 | 2.356 ± 0.163 | 7 | 0.200 |
| **Bandar Khayran** (Watanabe et al., 2017) | 59.5E ; 23.5N | 5 | 0.473 ± 0.060 | 11 | 1.606 ± 0.504 | 10 | 0.295 |
| **Galapagos** (Manzallo et al., 2014) | 92.5W ; 1.5S | 5 | 0.858 ± 0.283 | 5 | 7.288 ± 1.875 | 5 | 0.118 |
| **Saboga** (Mollica et al., 2018) | 78.5W ; 6.5N | 5 | 0.229 ± 0.148 | 18 | 2.341 ± 1.597 | 15 | 0.098 |

For the evaluation of measured coral calcification data (extension rate, bulk density and calcification rate), we used a

reference data set for *Porites* from Indo-Pacific reefs including 14 reefs from Hawaii (Grigg, 1981) and 29 reef sites from the Great Barrier Reef (Lough and Barnes, 2000). Each site represents mean annual calcification data of 6 – 15 coral records and annual mean SSTs (1903 – 1994, GOSTAPlus). For this dataset, we assume an oligotrophic growth habitat and calcification of these corals to be largely controlled by ambient SSTs, while an effect of nutrients and seawater acidification is negligible. The relationship between SST and the calcification parameters within these two regions was used to develop linear

calibrations for predictions of skeletal density, extension rate and calcification rate from ambient SST. Slopes and intercepts





of the calibrations slightly differ from those reported by Lough and Barnes (2000), because we confine the reference sites to Hawaii and the Great Barrier Reef. The resulting calibrations are as follows:

| | | | |
|---|---|---|---|
| Bulk density $[\text{g cm}^{-3}]$ | $= -0.1185 \times \text{SST} + 4.4158$ | $(r^2 = 0.49; p < 0.0001)$ | Eq. 1 |
| Extension rate $[\text{cm } yr^{-1}]$ | $= 0.3113 \times \text{SST} + 6.8994$ | $(r^2 = 0.90; p < 0.0001)$ | Eq. 2 |
| Calcification rate $[\text{g } cm^{-2} \, yr^{-1}]$ | $= 0.3342 \times \text{SST} + 7.1554$ | $(r^2 = 0.82; p < 0.0001)$ | Eq. 3 |

**2.1 Data matching and age model development**

Records of skeletal density and element concentrations were matched with optical microscope images allowing for the correlation of ablated spots from LA-ICP-MS with distinctive features on X-radiographs. Internal variations between the x-axis of the LA-ICP-MS record and the density records occur, because to some extent the LA-ICP-MS sampling paths were not ultimately straight due to following discrete corallites and avoiding bioerosions and incrustations. To overcome this, the chronologies of the density records inferred from straight transects orientated parallel to the direction of growth were applied
to the LA-ICP-MS records using AnalySeries software (Paillard et al., 1996).

Age models are based on Li/Mg-ratios in combination with Ba/Ca-ratios. Li/Mg is inversely related to temperature, which allows to identify the warm (inter-monsoon) and cool (monsoon) seasons (Harthone et al., 2013). In order to identify the upwelling season (SWM) among the two cool monsoons seasons SWM and NEM, we use Ba/Ca ratios as proxy (Tudhope et al., 1996) (Fig. 3). The individual coral records comprise three full years for coral 5.10 and five full years for coral 5.13 and
5.21, respectively. A detailed chronological frame for the Li/Mg records was established with the aid of the generalized annual record of remote sensing JPL MUR data (daily averaged  2003 – 2018) (Fig. 2), demonstrating average calendar dates of seasonal extremes to occur on 31.05. (SIM), 15.08. (SWM), 25.10. (AIM) and 06.02. (NEM). Average dates of inflection points between consecutive seasons from JPL MUR data were 18.03. (NEM – SIM), 25.06. (SIM – SWM), 30.09. (SWM – AIM) and 10.12. (AIM – NEM). This methodology allows tuning the age models to a total of eight tie-points per year (two
per season). Dates between tie-points were interpolated linearly and the entire time axis was resampled to monthly intervals using AnalySeries software (Paillard et al., 1996). Accuracy of the age model was checked by comparing the timing of seasonal remote sensing SST maxima and minima of individual years with dates of the generalized annual record (daily averages 2003 – 2018) and was found to be ± 4 weeks during NEM, ± 3 weeks during SIM and SWM, and ± 1.5 weeks during AIM.

**3 Results**

**3.1 SST calibration of Li/Mg records**

All multi-year monthly means of the Li/Mg records show pronounced patterns of two maxima and two minima within one annual cycle (Fig. 3). Li/Mg maxima coincide with peaks in Ba/Ca during summer upwelling (SWM) (Tudhope et al., 1996),





while a second, maximum of Li/Mg is reached during winter (NEM). Minima occur during the inter-monsoon seasons (SIM,

AIM). Monthly Li/Mg records exhibit the typical inverse relationship with remote sensing $SST_{rem}$ (JPL MUR) (5.10: $r^2$ = 0.65, p = 0.0016; 5.13: $r^2$ = 0.75, p = 0.0003; 5.21: $r^2$ = 0.91, p < 0.0001).

Multi coral monthly means in Li/Mg are used for the calibration of the Li/Mg thermometer with mean monthly $SSTs_{rem}$ (averages of 2003 – 2018) (Fig. 4). 83 % of the intra-annual multi coral monthly Li/Mg variation is explained by temperature and the resultant SST-calibration is estimated as:


$$\text{Li/Mg [mmol mol}^{-1}] \quad = -0.083 \, (\pm 0.012) \times SST + 4.029 \, (\pm 0.305) \qquad (r^2 = 0.83; p < 0.0001) \qquad \text{Eq. 4}$$

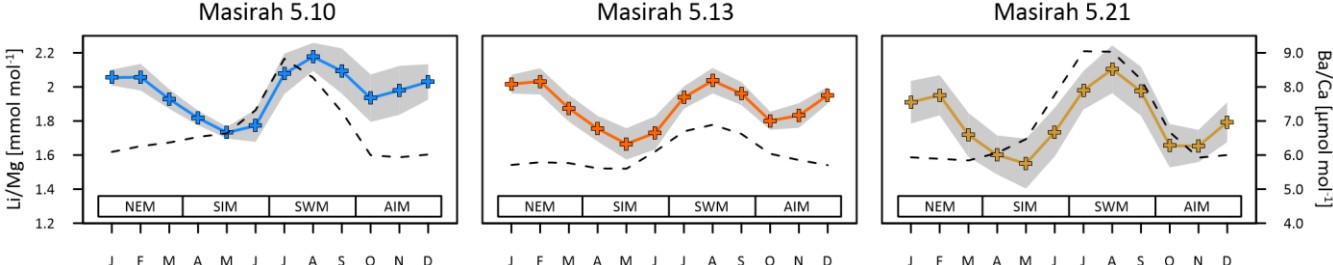

**Figure 3: Multi-year monthly means in Li/Mg of the three Masirah corals (blue, orange, yellow). The grey shaded area represent uncertainty (1σ) between equal months of consecutive years. Regular peaks in Ba/Ca (dotted line) associated to upwelling indicate**
**the southwest monsoonal season (Tudhope et al., 1996).**

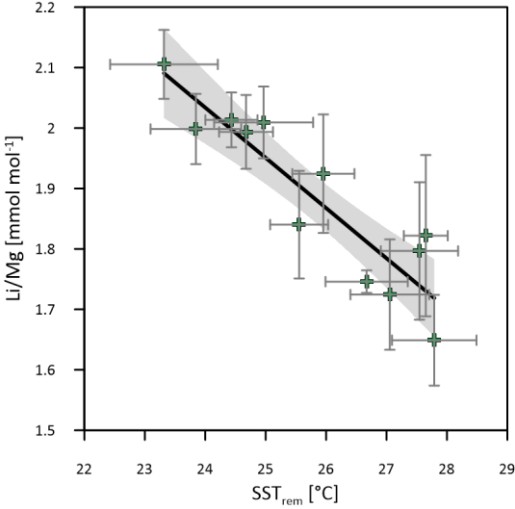

**Figure 4: Multi coral calibration of the Li/Mg thermometer with $SSTs_{rem}$ (monthly means 2003 – 2018, JPL MUR). Horizontal error bars (1σ) represent uncertainty between mean monthly SSTs of consecutive years (2003 – 2018) and vertical error bars**
**represent uncertainty of multi-year monthly mean Li/Mg ratios between the three Masirah corals. The grey shaded area indicates the 95 % confidence interval of the linear regression.**



## 3.2 Pattern of calcification

Mean annual bulk density of the coral samples is $1.02 \pm 0.02$ g cm$^{-3}$ (n = 3), $0.91 \pm 0.06$ g cm$^{-3}$ (n = 5) and $0.99 \pm 0.07$ g cm$^{-3}$ (n = 5) for coral 5.10, 5.13 and 5.21 respectively (Fig. 5). Two out of the three coral specimens (5.10 and 5.13) show a

pattern of two distinct high density bands (HDBs) between two bands of low density (LDB) within one annual growth increment. Skeletal portions of low density were deposited during SIM and AIM, high density portions formed during SWM and NEM. Coral sample 5.21 is similar to 5.10 and 5.13 but differs by lacking a well expressed LDB equivalent with AIM. Rather, it shows one wide LDB with density increasing from SWM to NEM. Mean annual extension rate of the coral specimens is $1.43 \pm 0.05$ cm yr$^{-1}$, $1.17 \pm 0.12$ cm yr$^{-1}$ and $1.53 \pm 0.11$ cm yr$^{-1}$ for coral 5.10, 5.13 and 5.21 respectively (Fig.

5). Corals 5.13 and 5.21 show three peaks of highest extension rates in March, June and October. Reduced linear growth occurs during SWM, NEM as well as in between April and May. Coral 5.10 slightly deviates from the pattern of the two other specimens due to the lack of elevated growth rates during June, leading to two peaks of maximum extension in February/March and October. In this specimen, a decrease in linear growth starts in SIM and reaches its minimum during SWM. Monthly mean calcification is mainly determined by linear extension rates, showing an equal inter-annual pattern for

all three corals studied. Net calcification rate is $1.45 \pm 0.08$ g cm$^{-2}$ yr$^{-1}$, $1.03 \pm 0.07$ g cm$^{-2}$ yr$^{-1}$ and $1.49 \pm 0.10$ g cm$^{-2}$ yr$^{-1}$ for coral 5.10, 5.13 and 5.21 respectively (Fig. 5).





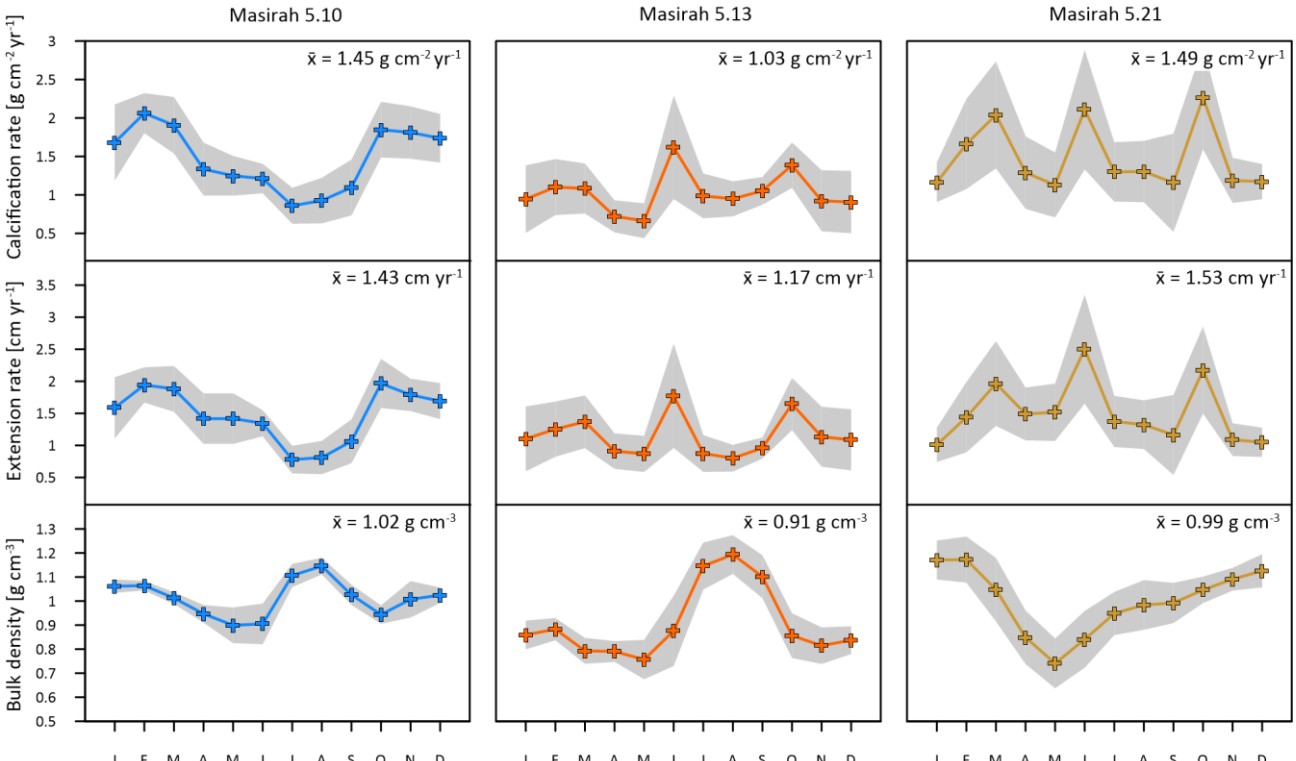

**Figure 5: Results of multi-year monthly means in skeletal density, extension rate and calcification rate of three corals from Masirah Island. The grey shaded area represent uncertainty (1σ) between equal months of consecutive years.**

## 4 Discussion

### 4.1 Li/Mg thermometry

SST reconstructions based upon the calibration of the Li/Mg thermometer described above (Eq. 4) reproduce the monthly curse of the $SST_{rem}$ data ($r^2 = 0.83$, $p < 0.0001$) as well as observed $SST_{in\text{-}situ}$ variations at Masirah Island ($r^2 = 0.93$, $p < 0.0001$) (Wilson, 2007). Temperature sensitivity of the Li/Mg – thermometer deduced from reconstructed seasonality is in good agreement with the majority of proxy calibration studies from the literature (Hathorne et al., 2013; Ross et al., 2019b; Cuny-Guirriec et al., 2019; Montagna et al., 2014; Fowell et al., 2016 (Forereef)). The intercept of the linear Li/Mg – SST calibration of the Masirah corals, however, is 4 – 8 °C higher than reported in the literature (Fig. 6). Analytical uncertainties that noticeably bias the Li/Mg ratio are unlikely as a source for high Li/Mg ratios, because systematic measurement discrepancies deduced from the JCp-1 QCM for Li/Ca and Mg/Ca would rather tend to underestimate the Li/Mg ratios (Li/Ca: +4.44 % ; Mg/Ca: +7.37 %; Table S1). In contrast, low seawater pH can cause high Li/Mg ratios by lowering the skeletal Mg content (Tanaka et al., 2015). A generally enhanced $Mg/Ca_{sw}$-variability found within regions affected by coastal upwelling is also assumed to cause biases in coral Li/Mg from undisturbed environments (Lebrato et al., 2020).





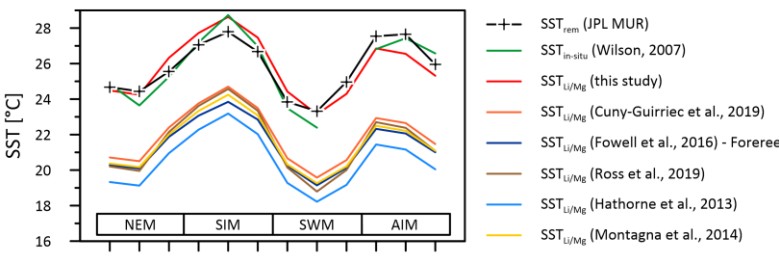


**Figure 6: Instrumental SSTs compared to multi coral monthly mean SSTs inferred from Li/Mg ratios of the Masirah corals using the calibration of this study (red line) and calibrations reported from the literature. Published Li/Mg – SST calibrations cause variable degrees of underestimation of coral growth temperatures at Masirah Island.**

### 4.2 Annual records of coral calcification

Annual bulk density and linear extension rate documented for Masirah *Porites* during this study reveal strong discrepancies to the reference data set of corals from Hawaii and the GBR (Lough and Barnes, 2000) (Fig. 7a-b). Multi coral annual mean skeletal density of the Masirah corals is 28 % lower, and mean annual linear extension rate is 20 % higher compared to the reference corals growing at equal temperatures at the GBR and Hawaii (Table 2). Unusually high extension rates were also reported from Omani reef sites at the Kuria Muria Islands (+44 %; Southern Arabian Sea, Oman; Montone, 2010) and

Bandar Khayran (+56 %; Gulf of Oman, Oman; Watanabe et al., 2017) (Fig. 1). Conversely, two corals from Marbat (southern Arabian Sea, Oman; Tudhope et al., 1996) show extension rates, which are in good agreement with the Indo-Pacific reference data (-2.8 %). Highly heterogeneous linear extension rates of massive *Porites* corals were also reported from the eastern equatorial Pacific upwelling zone. For instance, mean linear extension rate +60 % relative to western Pacific reef sites were reported from Galapagos Archipelago (Manzello et al., 2014), but -42 % from Saboga Island (Gulf of

Panama) (Mollica et al., 2018) (Fig. 7b). Thus, skeletal extension rates of corals from sites affected by upwelling are not a simple function of SST, neither within four spatially distinct coral regions from Oman nor at far distant reef sites of the eastern equatorial Pacific. In contrast, bulk density of reef corals from sites affected by seasonal upwelling is consistently lower than those reported for the reference data set (Galapagos: -38 % (Manzello et al., 2014); Saboga: -27 % (Mollica et al., 2018)) (Fig. 7a, Table 2).




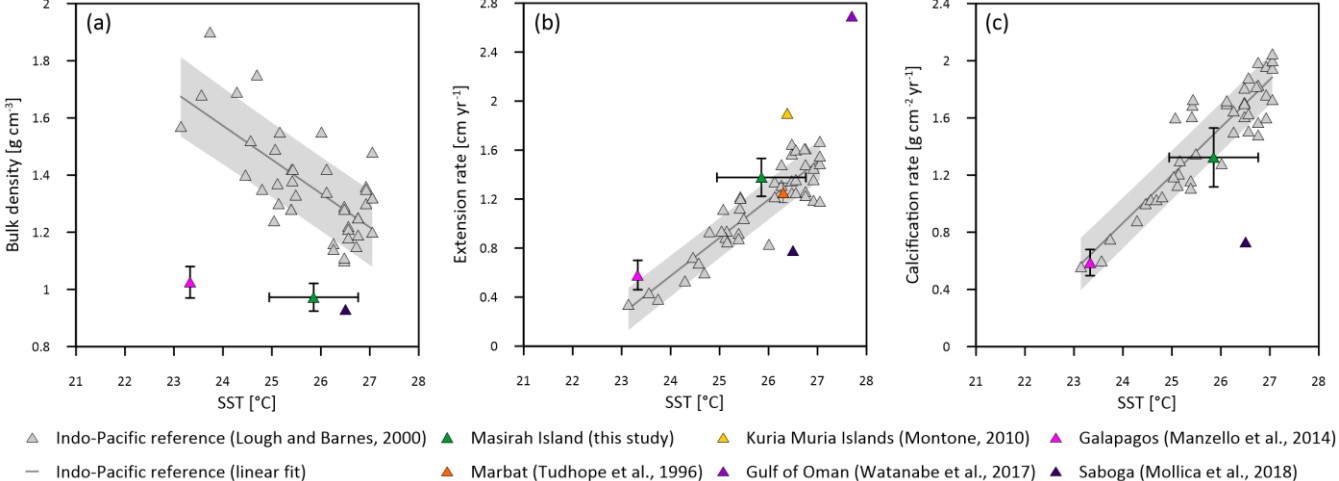

**Figure 7: Annual means in skeletal bulk density, linear extension rate and calcification rate versus sea surface temperature (SST) of corals from sites affected by intense seasonal upwelling (coloured) and reference corals from Hawaii and the GBR (grey). Error bars refer to variations in multi coral annual means (vertical) and in proxy derived SSTs. Linear regression and 95 % prediction interval (grey shaded) refer to the Indo-Pacific reference data (Lough & Barnes, 2000).**

A strong enhancement of coral linear extension rate, associated with a simultaneous loss in skeletal density was observed in response to nutrient fertilization (Dunn et al., 2012; Koop et al., 2001; Bucher and Harrison, 2001; Dodge and Brass, 1984, D'Olivo et al., 2013; Risk and Sammarco, 1991; Manzello et al., 2015). Nonetheless, published records reveal no correlation between annual extension rate and nutrients, neither $NO_3^-$ nor $PO_4^{3-}$ at coral sites are affected by seasonal upwelling ($r^2$ ($NO_3^-$) = 0.26, p = 0.30 ; $r^2$ ($PO_4^{3-}$) = 0.05, p = 0.68) (Table 1). In contrast, however, extension rate is strongly correlated with the $PO_4^{3-}/NO_3^-$-ratio (P/N) ($r^2$ = 0.91, p = 0.003) rather than with annual SST ($r^2$ = 0.57, p = 0.085) when seasonal upwelling occurs (Fig. 8a,b). When extension rates are expressed as the deviation from the values predicted by the reference corals (Eq. 2), the residuals still were well related to P/N ($r^2$ = 0.71, p = 0.036) (Fig. 8c). However, the quality of correlation is strongly determined by the data of the *Porites* from Saboga (Mollica et al., 2018). This is because the extremely low extension rate at relatively high SST downgrade the positive relationship apparent from the data of the remaining corals from upwelling sites (Fig. 8a). An imbalanced P/N with disproportionally high $NO_3^-$ within the Gulf of Panama (Table 1) likely inhibits linear growth, outweighing the stimulating effect of high SSTs on extension rate (Koop et al., 2001). Diminished extension rate within the Gulf of Panama compared to sites unaffected by upwelling is also reported for *Pocillopora damicornis* (Glynn, 1977; Manzello, 2010). *Pavona clavus*, however, shows enhanced growth rates within the Gulf of Panama compared to sites unaffected by upwelling, while growth rates of *Pavona gigantea* are indistinguishable (Manzello, 2010; Wellington and Glynn, 1983). Variable coral growth rates at one and the same site might either demonstrate (1) genus/species related differences of growth sensitivities to ambient P/N, (2) local heterogeneities in nutrient distribution over intra-reef scales or (3) variable conditions for linear growth, e.g. light penetration (Omata et al., 2008).




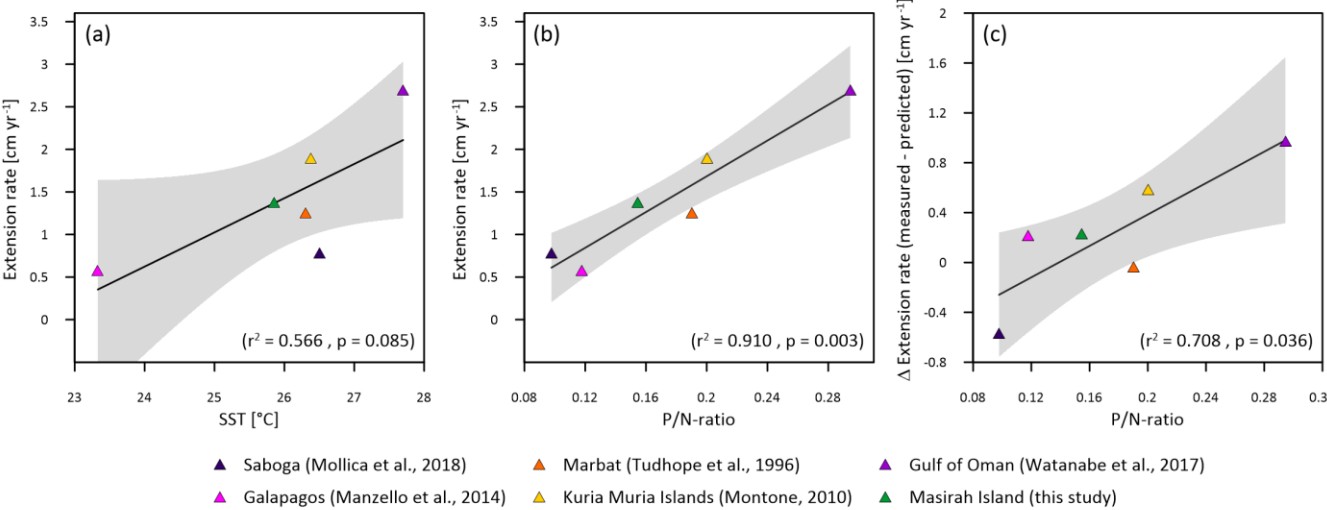

**Figure 8: Mean annual extension rates of *Porites* from upwelling regions as a function of (a) SST, (b) P/N. Deviations between measured extension rates and those predicted by the reference corals (Lough and Barnes, 2000) as function P/N (c). The grey shaded area represents the 95 % confidence interval of the regressions (solid black line).**


Published data of bulk density from sites affected by strong seasonal upwelling are extremely sparse (n = 2). Including this study, a total number of three site averaged data values on bulk density brought up no unambiguous correlations with either SST, $NO_3^-$, $PO_4^{3-}$ or P/N. Mollica et al. (2018) showed, however, that low skeletal density of the coral from the Panama upwelling zone (Saboga) is caused by a diminished aragonite saturation state of the calcifying fluid ($\Omega_{cf}$) compared to corals

from sites unaffected by upwelling. Hence, an equal underlying mechanism can also be assumed to account for low skeletal density of corals from Masirah Island and Galapagos.

Assuming annual extension rates being in accordance to those of the reference corals, diminished skeletal density of the Masirah and Galapagos corals would result in net calcification rates of -24.8 % ($\Delta$Calc = -0.37 g cm$^{-2}$ yr$^{-1}$) and -42.1 % ($\Delta$Calc = -0.27 g cm$^{-2}$ yr$^{-1}$), respectively. However, enhanced extension rates at Masirah and Galapagos attenuate this effect,

resulting in calcification rates only 10.86 % and 7.32 %, respectively, lower than that of the reference corals (Table 2). This compensating effect of high extension rates on calcification rates is not present in the *Porites* from Saboga. Rather, low extension rates amplify the negative effect of low density on calcification rate. This demonstrates the stimulating (inhibiting) effect of a high (low) P/N on extension rate also significantly determines the calcification rate.






**Table 2: Comparison of site-specific measured annual mean calcification parameters (skeletal density, extension rate, calcification rate) of corals affected by strong seasonal upwelling to values predicted for annual SSTs by the western Pacific reference data set (Hawaii and GBR) (Lough and Barnes, 2000). Empty cells indicate the lack of measured data. Galapagos calcification data involves the sites Devil's Crown and Urvina Bay (Manzello et al., 2014).**

| | n | SST [°C] | Measured values | | | Predicted values | | | Δ (measured – predicted) | | | | | |
|---|---|---|---|---|---|---|---|---|---|---|---|---|---|---|
| | | | Bulk den. [g cm⁻³] | Ext. rate [cm yr⁻¹] | Calc. rate [g cm⁻² yr⁻¹] | Bulk den. [g cm⁻³] | Ext. rate [cm yr⁻¹] | Calc. rate [g cm⁻² yr⁻¹] | Δ Bulk den. [g cm⁻³] | [%] | Δ Ext. rate [cm yr⁻¹] | [%] | Δ Calc. rate [g cm⁻² yr⁻¹] | [%] |
| **Masirah Island** (this study) | 3 | 25.85 ± 0.91 | 0.97 ± 0.05 | 1.38 ± 0.15 | 1.32 ± 0.21 | 1.35 | 1.15 | 1.48 | -0.38 | -28.08 | 0.228 | 19.86 | -0.16 | -10.86 |
| **Marbat** (Tudhope et al., 1996) | 2 | 26.30 | - | 1.25 | - | - | 1.29 | - | - | - | -0.04 | -2.82 | - | - |
| **Kuria Muria** (Montone, 2010) | 1 | 26.38 | - | 1.90 | - | - | 1.31 | - | - | - | 0.59 | 44.74 | - | - |
| **Bandar Khayran** (Watanabe et al., 2017) | 1 | 27.70 | - | 2.70 | - | - | 1.72 | - | - | - | 0.97 | 56.42 | - | - |
| **Galapagos** (Manzello et al., 2014) | 2 | 23.33 ± 0.00 | 1.03 ± 0.06 | 0.58 ± 0.12 | 0.59 ± 0.09 | 1.65 | 0.36 | 0.64 | -0.63 | -38.18 | 0.22 | 59.68 | -0.05 | -7.32 |
| **Saboga** (Mollica et al., 2018) | 1 | 26.50 | 0.93 | 0.78 | 0.73 | 1,28 | 1.35 | 1.70 | -0.35 | -27.34 | -0.57 | -42.22 | -0.98 | -57.35 |


## 4.3 Monthly records of coral calcification

Multi coral mean monthly calcification data of the three corals from Masirah Island demonstrate that the deficit of skeletal density found within the annually resolved data exists also over the entire year and co-varies consistently with mean monthly SST ($r^2 = 0.95$, $p < 0.0001$ ) while no link is apparent with a certain season, i.e. the upwelling (Fig. 9a). Nonetheless, the

largest difference in mean monthly skeletal density to values predicted by the reference data occur during the monsoon seasons ($\Delta\rho_{SWM}$ = -0.51 ± 0.05 g cm⁻³; $\Delta\rho_{NEM}$ = -0.44 ± 0.07 g cm⁻³), the lowest differences during the inter-monsoon seasons ($\Delta\rho_{SIM}$ = -0.26 ± 0.03 g cm⁻³; $\Delta\rho_{AIM}$ = -0.33 ± 0.06 g cm⁻³). Excessively high nutrient concentrations and low $\Omega_{sw}$ hardly explain year-round low skeletal density, because the nutrient pulse and simultaneously diminished $\Omega_{sw}$ are temporally limited to the SWM (Fig. 2). Year-round low skeletal density is therefore rather related to permanently low aragonite

saturation at the site of calcification ($\Omega_{cf}$) (Mollica et al., 2018). A similar pattern of relatively constant, but low $\Omega_{cf}$ over the entire year, independent from external variations in $\Omega_{sw}$ caused by upwelling, was reported also for the Saboga coral (N. Mollica, personal communication, 2021). This finding would imply that there is no intensified upregulation of internal $\Omega_{cf}$ relative to $\Omega_{sw}$ during the non-upwelling seasons and a rather low level of internal $\Omega_{cf}$ is maintained over the entire year (McCulloch et al., 2017; DeCarlo et al., 2018; D'Olivo and McCulloch, 2017). Since the non-upwelling season in the

Arabian Sea lasts for approximately nine months, coral growth occurs at $\Omega_{sw}$ of 3.5 – 4 for most of the time (Omer, 2010), which is above values assumed to impede skeletal calcification (Kleypas et al., 1999). Similar $\Omega_{sw}$ are reported also from several undisturbed Pacific coral reefs which are unaffected by upwelling, though internal $\Omega_{cf}$ of these corals is approximately 25 % higher compared to those being affected by upwelling (Mollica et al., 2018). As an explanation, we propose that internal upregulation processes of corals from Masirah Island are not capable to adapt completely to the ocean

chemistry change on a quarterly scale, therefore maintaining a low gradient between $\Omega_{sw}$ and $\Omega_{cf}$ in order to survive the more harsh condition during SWM.





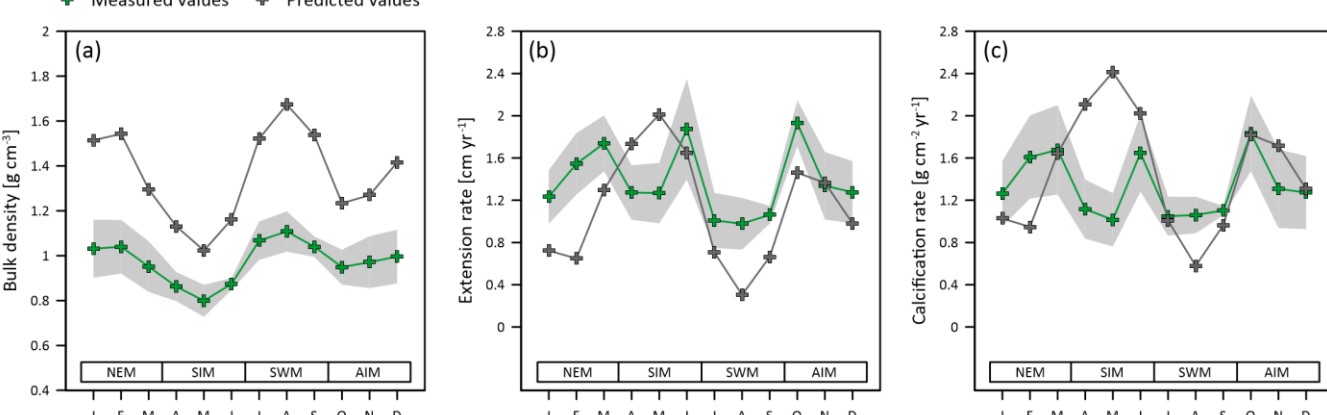

**Figure 9: Multi coral monthly means in skeletal density, extension rate and calcification rate of the Masirah corals (green) and**
**values predicted based on SSTs (black) by the Indo-Pacific reference data (Lough and Barnes, 2000). The grey shaded area**
**represents variations (1σ) between multi-year monthly means of the three Masirah corals.**

Multi coral mean monthly extension rate is only weakly positively correlated with SST ($r^2$ = 0.26, p = 0.094), which is
caused by noticeably small extension rates during April and May despite of high SIM temperatures (Fig. 9b). SIM
constitutes the only season, when extension rates fall below values predicted by the western Pacific reference coral dataset
($\Delta Ext_{SIM}$ = − 0.33 ± 0.41 cm yr$^{-1}$). Interestingly, coral spawning at Omani reef sites is reported to take place between March
and May (Howells et al., 2014). Correspondingly, a reduction of extension rate during SIM could be linked to high-energy
expenditures required for reproduction (Cabral-Tena et al., 2013). Irrespective of low extension rates during April and May,
the monthly variation of extension rate during the remaining annual period is strong positively related to SSTs ($r^2$ = 0.67, p =
0.0038). Best accordance between measured extension rates and values predicted by western Pacific corals occur during
AIM ($\Delta Ext_{AIM}$ = 0.24 ± 0.20 cm yr$^{-1}$). In contrast, skeletal extension during the monsoon season is remarkably higher than
predicted for ambient SSTs ($\Delta Ext_{SWM}$: 0.46 ± 0.16 cm yr$^{-1}$ ; $\Delta Ext_{NEM}$: 0.62 ± 0.20 cm yr$^{-1}$). While the nutrient pulse
associated with upwelling would only affect SWM extension rates (Fig. 2), constantly high levels of nutrients and/or a
constantly high P/N during the entire year are potentially to buffer the negative effect of low temperatures on extension rates
during NEM and SWM (Dunn et al., 2012; Koop et al., 2001; Smith, 2001). Here, we infer a mechanism in which nutrients
promote biochemical processes keeping efficiency of symbiont photosynthesis upright despite of low temperatures, thus
allowing for the maintenance of a relatively thick tissue layer. The tissue layer thickness is supposed to control monthly
upward growth of the coral (DeCarlo and Cohen, 2017; Godinot et al., 2011). It has been shown from coral growths at the
upper temperature threshold that the availability of nutrients, in particular a high P/N increases the resilience to bleaching by
maintaining photosynthesis upright (Ezzat et al., 2016; Riegl et al., 2019; Rosset et al., 2017; Wiedemann et al., 2013).
Hence, the decline of the tissue layer thickness is less expressed leading to smaller decreases in monthly extension rate





(Marangoni et al. 2021; DeCarlo and Cohen, 2017). We assume an equal mechanism to account for sustaining relatively high extension rates of the Masirah corals at the lower temperature limits, in particular because P/N in northern Arabian Sea is high (Kleypas et al., 1999). A comparable mechanism of nutrient stimulated tissue growth and a concomitant increase in

linear growth was also proposed for *Pavona clavus* from the Gulf of Panama (Wellington and Glynn, 1983).

Multi coral mean monthly calcification rate inferred from data of this study is essentially controlled by the effects of extension rate ($r^2 = 0.88$, $p < 0.0001$) rather than skeletal density ($r^2 = 0.016$, $p = 0.70$). Calcification rate during SIM shows the highest deficits compared to western Pacific reference corals ($\Delta Calc_{SIM} = -0.92 \pm 0.42$ g cm$^{-2}$ yr$^{-1}$) because both, extension rate and skeletal density are low at the same time. Best accordance to values predicted from the reference data set

is documented for the AIM ($\Delta Calc_{AIM} = -0.14 \pm 0.19$ g cm$^{-2}$ yr$^{-1}$). During both monsoon seasons, calcification is higher than of corals from the reference data set, because elevated extension rates overcompensate for diminished skeletal density ($\Delta Calc_{NEM} = 0.31 \pm 0.26$ g cm$^{-2}$ yr$^{-1}$; $\Delta Calc_{SWM} = 0.22 \pm 0.19$ g cm$^{-2}$ yr$^{-1}$).

### 4.4 Prospective view on coral calcification under low $\Omega_{sw}$ and eutrophic conditions

The results of this study show, that seawater nutrients have potential to attenuate the negative effect of ocean acidification on

reef coral calcification. This is because nutrient-stimulated linear extension rates can compensate the negative effect of low $\Omega_{sw}$ driven decrease in skeletal bulk density. However, while a low $\Omega_{sw}$ environment has an unconditionally detrimental effect on reef coral bulk density, the positive effect of eutrophication on linear extension rate highly depends on optimal nutrient conditions i.e. the concentrations and balance between the essential nutrients e.g. P/N. As shown by the coral from Saboga, low P/N has a negative effect on extension rate, which amplifies the negative effect of low skeletal density on net

calcification. In contrast, high P/N in the northern Arabian Sea upwelling zone stimulates linear extension rates, which compensate the loss in skeletal bulk density and thus enable to maintain relatively high calcification rates. On an intra-annual basis, we found periods during which increased extension of the Masirah corals (Arabian Sea) fully compensate and even overcompensate for the diminished skeletal bulk density.

In addition to upwelling regions being heralded as refuges for tropical coral species in times of rising SST (Riegl & Piller,

2002; Chollett et al., 2010), patterns of reef coral calcification from these regions provide further implications to the future evolution of coral reefs under low pH and eutrophic conditions. The here presented data suggest optimal nutrient environments (e.g. a high P/N) to have high potential to compensate the negative effect of ocean acidification on reef coral calcification. In this case, a skeletal thickening strategy, which is present in typical modern tropical coral reefs, will change into an extension strategy (Carricart-Ganivet, 2004). A relatively high net calcification maintained in this process is

beneficial to the positive balance of carbonate accumulation on tropical coral reefs, thereby preserving the fundamental prerequisite for the reef habitat and the health of the associated ecosystem. The identification of valuable refuges with particular significance for the conservation of tropical coral reefs should therefore consider not only SST but also nutrient conditions.

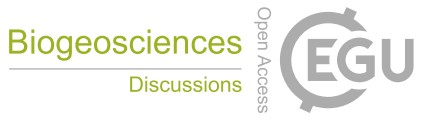

## 5 Conclusion

This study documents patterns of skeletal calcification in corals (*Porites*) from the Arabian Sea upwelling zone (Masirah Island, Oman). Compared to corals from typical reef sites unaffected by upwelling used as a reference dataset (Hawaii and Great Barrier Reef), mean annual skeletal density is reduced by 28 %, which is in good agreement with values reported from corals of eastern Pacific upwelling zones. The intra-annual variability of skeletal density is negatively related to SST, but skeletal density remains consistently too low throughout the entire year relative to the reference data set, likely because of a

comparatively low aragonite saturation state at the site of calcification ($\Omega_{cf}$). $\Omega_{cf}$ is assumed to be kept low by the coral, but relatively constant and independent of changes in external $\Omega_{sw}$ between the upwelling and non-upwelling seasons.

Mean annual extension rate is 20 % higher than for the reference data set. In particular, extension rates are greatly enhanced during the cool monsoon seasons, presumably due to a permanently high P/N contributing to the maintenance of a thick tissue layer that promotes rapid upward growth of the skeleton. The stimulating effect of nutrients was not observed during

the two warm inter-monsoon seasons, despite a year-round high P/N in the Arabian Sea. However, while extension rate is compatible with the western Pacific reference data during autumn inter-monsoon, it is noticeably diminished during spring inter-monsoon, likely corresponding with spawning cycles.

Enhanced annual extension rate result in the negative effect of low density on the calcification rate being attenuated from -25 % to -11 %. As intra-annual calcification is also strongly controlled by extension, calcification rates during the inter-

monsoon seasons are lower and during the monsoons higher than those of the reference corals.

As a conclusion of this study, we recommend more consideration to be spend to local nutrient conditions when identifying suitable refuges for tropical coral reefs. Optimal nutrient environments (e.g. a high P/N) have significantly higher potential to compensate the negative effect of ocean acidification on reef coral calcification. Adequate rates of carbonate accumulation are the fundamental prerequisite to the preservation of reef habitats and enable the persistence of this unique ecosystem.

**Authors contribution**

P. M. Spreter designed this research in close collaboration with T. C. Brachert and M. Reuter. Field work was carried out by P.M. Spreter, T.C. Brachert, M. Reuter and substantially supported by O. Taylor. Laboratory analyses were performed by P. M. Spreter and R. Mertz-Kraus. All authors had a contribution in writing and improving the manuscript.

**Competing interests**

The authors declare that they have no conflict of interest.





**Acknowledgements**

We are greatful to the Ministry of Environment and Climate Affairs (Muscat, Oman) for approving sampling and export permission (Permit numbers: 6210/10/44 and 177/2018). We kindly thank M.A. Claereboudt (Sultan Qaboos University, Muscat, Oman) for his support. Funding by the Deutsche Forschungsgemeinschaft (DFG, grant BR 1153/20-1, -2 and ME
3761/4-1) is gratefully acknowledged.

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
