# Peer review of "Calcification response of reef corals to seasonal upwelling in the northern Arabian Sea (Masirah Island, Oman)"

_Biogeosciences, 2021_

## Author Comment (AC1)

Legend:
- Indo-Pacific (from Lough and Barnes, 2000)
- Masirah Island (all samples available)
- Masirah Island (mean all samples available)
- Masirah Island (samples presented in manuscript)

---

## Author Comment (AC2)

Authors' response to reviews of

**"Nutrients attenuate the negative effect of ocean acidification on reef coral calcification in the Arabian Sea upwelling zone (Masirah Island, Oman)"**

Philipp M. Spreter et al.

Biogeosciences, doi.org/10.5194/bg-2021-213-RC2, 2021

RC: Reviewers' comment

AR: Authors' response

**1. Introduction:**

on behalf of my co-authors, I would like to thank the reviewers for commenting on our manuscript. Their supporting comments and ideas are important for us to improve the significance of our work.

Given that both reviewers stated fundamental concerns about the framing and interpretation of our data, I would like to address these general aspects in advance, before discussing the single reviewer comments in detail.

The reviewers express general doubts about the reliability of the mean annual calcification data (extension rate and bulk density) of *Porites* from Masirah Island reported by us because of a comparatively small number of replicates (n=3). Accordingly, low mean annular skeletal bulk density and enhanced linear extension rate of *Porites* from Masirah Island recorded by our study relative to those of *Porites* from the Indo-Pacific is suggested to be a possible artefact of a small database. However, a larger number of specimens and respective calcification data is available to us. This supplementary information was provided to the open discussion in a reply to RC1 (16 Sept 2021). The dataset clearly shows that the calcification data presented in our discussion paper are not biased by poor statistics, but document a mean situation for the region. We selected three out of the six samples for further geochemical analysis (laser ICP-MS) that were considered representative for the site studied. These geochemical analyses enabled us to obtain monthly resolved calcification patterns of *Porites* from a region affected by intense seasonal upwelling, which is indeed a novel approach. We will therefore keep the focus on these three samples also in the revised version of the manuscript. However, as no *Porites* growth data exists from the Arabian Sea, the entire data of all 6 specimen will be added to the revision as an electronic supplement. In this way, we hope to encourage further *Porites* growth studies in a hitherto unexplored region of the western Indian Ocean.

Being aware that the number of three samples is statistically problematic, we will re-arrange the manuscript as suggested by both reviewers and place the focus more on a combination of the proxy data (e.g. Li/Mg, Ba/Ca) and intra-annual growth patterns presented in the original manuscript. To investigate the influence of upwelling on coral calcification, we discuss differences between the seasonal growth patterns within our samples, in particular those observed during the upwelling and non-upwelling seasons. By doing so, "intra-reef variabilities" in coral growth problematized by Reviewer 2 are largely negligible, as local factors and the genotype remain constant within a single specimen. We strongly feel discussing the existing data in this way is scientifically more robust and solves any legitimate concerns of the reviewers.

**2. Reviewer # 1 (RC1):**

- **2.1.** "Three replicate cores is just not enough to have robust statistics...Actually, the only way I see forward for the authors (besides getting more corals) is to completely re-frame the study based on the geochemistry."
- AR: This comment is addressed in detail in the introduction.
- 2.2. "I had no idea there was going to be Li/Mg geochemistry in this study based on the title and abstract."
- AR: Information on the application of Li/Mg and Ba/Ca trace element analyses for the interpretation of intraannual patterns of coral calcification will be provided in the revised version of the abstract.
- **2.3.** "I don't think it makes sense to compare the growth rates to just the GBR."
- AR: In the discussion paper, we compare our growth data from Masirah Island with reference data from the GBR (29 sites, 245 colonies) and from Hawaii (14 sites, 140 colonies) (Grigg, 1981; Lough and Barnes,

2000). These two datasets represent the largest of their kind for Porites. The calcification variables (skeletal density, extension rate and calcification rate) are all significantly correlated with SST (Lough and Barnes, 2000). For this reason, the regressions of these relationships are frequently used as "baselines" for estimating the performance of recent and fossil Porites with regard to their bulk density, extension rates and calcification rate at a certain SST (Brachert et al., 2016; Brachert et al., 2020; Lough et al., 2016; Manzello et al., 2014). However, we agree that our approach of applying the regressions from the Indo-Pacific reference dataset to predict calcification performance of *Porites* from the Arabian Sea possibly overstresses the global validity of the causal relationship between SST and calcification because of a small sample number of specimens in our dataset (n = 3). For this reason, in our revised manuscript, we refrain from predicting "theoretical values" for the calcification variables from a given SST and instead provide a more qualitative comparison of our data with published Porites growth data. In this way, we will provide a more global perspective, by showing additional (published) Porites growth data from Thailand, Western Australia, the Gulf of Mexico plus three other regions affected by upwelling, in addition to the previously presented data from the GBR and Hawaii. All published records from globally distributed upwelling sites show a conspicuous and consistent pattern of low skeletal density without any exception (Fig.1).

**Figure 1:** Annual means in *Porites* skeletal bulk density versus annual sea surface temperature (SST) from various Indo-Pacific and Western Atlantic reef sites. Note that all reef sites affected by seasonal upwelling show exceptionally low skeletal bulk density.

**3. Reviewer # 2 (RC2):**

**Concerns:**

- **3.01.** "Overall, an n of 3 is fairly small to make these growth assumptions with... growth is highly variable across locations and genotypes and thus would require a higher n to make stronger predictions/estimates."
- AR: This comment is addressed in detail in the introduction.
- **3.02.** "Coral growth is reliant on a lot of different biological processes that are largely ignored throughout this text. While calcifying fluid is discussed, the authors never actually make these measurements. Further, the authors ignore how much variability in coral growth there is within a genus or even a species and over-estimate the reliance of growth on SST. The authors should consider what other factors can impact the variability across their cores and work to put it more in the context of the biology of corals. Along these same lines, coral cores collected from the Pacific may not experience similar environmental conditions and may likely have different populations/species of Porites corals, thus comparison of the corals from this study with Pacific corals should be made with caution."
- AR: This reviewer comment will be split into three sections, which will be answered separately in the following:

"Coral growth is reliant on a lot of different biological processes that are largely ignored throughout this text."

AR: We are working on dead coral material and do not want to speculate about the biological processes in detail, but rather discuss coral calcification with respect to environmental controls. In this context, we strongly feel to address fundamental environmental factors such as SST, pH,  $\Omega_{Arag}$  and nutrients which have been shown to have the greatest impact on reef coral calcification (Courtney et al., 2017; Ross et al., 2019).

"While calcifying fluid is discussed, the authors never actually make these measurements."

AR: Indeed, we did not perform combined analyses of B/Ca and  $\delta^{11}$ B on our specimens in order to estimate the  $\Omega_{cf}$  of the calcification fluid. Recent publications, however, have led us to consider  $\Omega_{cf}$  as a major parameter controlling skeletal density (Mollica et al., 2018). Considering this, it is essential for us to discuss the variability in skeletal density with regard to possible influences of  $\Omega_{cf}$ . Systematic measurements of B/Ca and  $\delta^{11}$ B are currently underway in the frame of a cooperation beyond the group of co-authors for this manuscript.

"Further, the authors ignore how much variability in coral growth there is within a genus or even a species and over-estimate the reliance of growth on SST. Along these same lines, coral cores collected from the Pacific may not experience similar environmental conditions and may likely have different populations/species of Porites corals, thus comparison of the corals from this study with Pacific corals should be made with caution"

- AR: An overwhelming number of studies demonstrate SST to have the most significant impact on reef coral calcification (Courtney et al., 2017; Lough and Barnes, 2000; Lough, 2008; Lough et al., 2016). For this reason, we do not agree with the statement of having overestimated the reliance of growth on SST. However, as we already mentioned in our response to RC1-2.3, we agree that our approach of applying the regressions from the Indo-Pacific reference dataset to predict calcification performance of *Porites* from the Arabian Sea likely poses problems. This is because of a small number of replicates (n = 3), which means that we are likely unable to capture possible full growth variability at Masirah Island. To overcome this, we decided to refrain from calculating theoretical calcification based on the regressions from Indo-Pacific *Porites* but rather to focus on the intra-annual growth variability within a single specimen is now being discussed in terms of the coral's calcification response to seasonally changing environmental factors such as SST,  $\Omega_{Arag}$ , light availability for photosynthesis and seawater nutrients, while growth variability across our samples is also discussed with regard to locational- and inter-species variability.
- **3.03.** *"While the satellite SST matched very well with in situ SST measurements, the other satellite phosphate and nitrate parameters were not similarly grounded in truth. Further, the lower resolution for these satellite products is concerning since nutrient values can be highly variable across a spatial gradient."*
- AR: WOA18 nutrient data is not a satellite product, but a generalized interpolation of all available in-situ observations performed within each 1° square (Garcia et al., 2018). Due to reframing of our study with a new focus on intra-annual growth patterns, we only consider monthly resolved WOA18 nutrient data from Masirah Island in the revised version of the manuscript (Fig.2, discussion paper). These data show relatively homogeneous nutrient concentrations during the non-upwelling season and a pronounced maximum of both essential nutrients (phosphate, nitrate) during the upwelling season. This intra-annual dynamic of nutrient concentrations is generally associated with upwelling areas and has also been observed in the northern Arabian Sea through in-situ observations (Currie et al., 1973; Savidge et al., 1990). In combination with the excellent agreement to skeletal Ba/Ca records, we assume that the general pattern of monthly resolved WOA18 nutrient concentrations at Masirah Island qualitatively reflect the intra-annual nutrient cycle of surface waters (Lea et al., 1989; Montaggioni et al., 2005; Tudhope et al., 1996). Nevertheless, we are aware of temporal and spatial heterogeneities in nutrient distribution and discuss this based on variabilities in the Ba/Ca records across specimen.
- **3.04.** "Finally, because of the discussion of ocean acidification is a major component of this manuscript, it also may be valuable for the authors to also include carbonate parameters for the area (i.e., omega-aragonite, TA, pH, etc.)."

AR: We regret that except for data from the Arabian Sea Expedition (1995) recorded during the southwest monsoon (upwelling) and northeast monsoon (non-upwelling), no additional information on the carbonate chemistry of the northern Arabian Sea is available. Existing data from the Arabian Sea Expedition on seawater aragonite saturation ( $\Omega_{sw}$ ) are reported in the methods section of the discussion paper (lines 101-105) and are discussed in the following (line 370).

**Minor edits:**

**Abstract**

- **3.05.** "Lines 14-15: This statement is a bit of an overstatement. Calcification responses to changing environments is very well studied. If this is intended to be in terms of a specific type of environmental variation, then that needs to be clarified."
- AR: We have revised the sentence to read: "The calcification response of reef corals to rapid and simultaneous changes in  $\Omega_{sw}$  and seawater nutrient concentrations is not well studied, however."

**Introduction**

- 3.06. "Line 29: Consider replacing 'zooxanthellate' with symbiotic."
- AR: We have replaced the word "zooxanthellate" by "symbiotic".
- **3.07.** "Lines 36-37: This statement should be backed with the literature. Coral calcification is not highly debated; however, responses are highly variable. I recommend referencing several papers covering this here."
- AR: We have revised the sentence to read: "The responses of reef coral calcification on this rapidly changing environment are highly variable and remain currently a matter of debate." and covered this statement with literature from Cornwall et al., (2021), Guan et al., (2020) and Hall et al., (2018).
- **3.08.** "Lines 50-52: This statement seems misplaced and should be incorporated better within the introduction. Additionally, recent reviews suggest different calcification responses in corals under global change (see Cornwall et al 2021, Global declines in coral reef calcium carbonate production under ocean acidification and warming, PNAS)."
- AR: We have removed this sentence because variable responses in corals under global change are already addressed in the introduction where the reference to the work of Cornwall et al., (2021) is cited (see RC2-3.07).
- 3.09. "Line 54: savage disposal? Do you mean sewage?"
- AR: We have revised the spelling mistake to "sewage".
- 3.10. "Lines 54-55: This is an incomplete statement."
- AR: We have revised the sentence to read: "Eutrophication can have both beneficial as well as detrimental effects on coral growth, however (Tomascik and Sander, 1985; Tomascik, 1990)."
- **3.11.** "Lines 55-56: Again, I suggest updating your language here to reflect more recent terminology of coral algal symbionts (see LaJeunesse et al. 2018, Systematic Revision of Symbiodiniaceae Highlights the Antiquity and Diversity of Coral Endosymbionts, Current Biology)."

- AR: We have revised the sentence to read: "Reef corals are highly adapted to oligotrophic waters with microalgae symbionts to allow an efficient use of essential nutrients and to outcompete other fast-growing biota on a reef whose growth is inhibited by the undersupply of nutrients (Barrot and Rohwer, 2012; LaJeunesse et al., 2018; Vermeij et al., 2010)."
- **3.12.** "Lines 66-68: These sentences could be a bit stronger to introduce this important topic in your introduction."
- AR: We have revised the sentence to read: "Understanding how coral calcification responds to rapid changes in seawater nutrient conditions and  $\Omega_{sw}$  is critical for more accurate predictions on the persistence of reef habitats under the influence of global change."
- **3.13.** *"Lines 70-72: This statement would benefit from a clear connection of how calcification responses from upwelling locations can be applied to systems without upwelling."*
- AR: We have revised the sentence to read: "This allows these regions to serve as natural laboratories to investigate the calcification response of reef corals to these multiple environmental stressors that are likely to affect global coral reefs in the near future (Camp et al., 2018; Wizemann et al., 2018)."
- **3.14.** "Lines 84-125: This section should be moved into the methods as a section describing the sites. A condensed version of this could be included in the previous paragraph to describe the sites if that is desired."
- AR: We have moved the chapter "Arabian Sea climate and oceanography" to the beginning of the methodology section in the revised version of the manuscript.

**Methods**

- **3.15.** *"Lines 128-134: It sounds like these colonies were not collected in situ, rather collected as dead skeletons from the beach. What about differences in local conditions? If these corals were washed up on shore you don't really know what depth or location they came from? Also what years? How do you know when they washed up on the beach?"*
- AR: This reviewer comment will be split into three sections, which will be answered separately in the following:

"...when were they washed up on the beach?"

AR: Certainly, all the corals were washed up on the beach by the same event, as they were all collected from the same storm deposit, which is attributed to Cyclone Gonu in 2007 (Fritz et al., 2010).

"...also what years?"

AR: The detailed information about the calendar years represented in our records is marginal, as we establish generalized annual calcification records with monthly resolution from several years of the total record lengths (Multi-year monthly means). These generalised records represent the mean monthly calcification of *Porites* corals at Masirah Island during the more recent current era.

"...What about differences in local conditions...depth or location?"

- AR: In the revised version of our manuscript, we discuss intra-annual variability of calcification (in particular between the upwelling season and non-upwelling season) with regard to seasonal environmental changes. Location factors are hereby to be neglected, as they remain uniform within a single specimen.
- **3.16.** "Lines 216-219: Unclear what these numbers and acronyms represent. Please rephrase in a clearer way."

AR: These acronyms refer to the two monsoon seasons (SWM = Southwest Monsoon, NEM = Northeast Monsoon) and the intermonsoon seasons (SIM = Spring Intermonsoon, AIM = Autumn Intermonsoon), respectively, and are introduced in the preceding chapter "Arabian Sea climate and oceanography" (lines 88-92) of the discussion paper. However, to ensure better reading in the section "Data matching and age model development" (lines 216-216), we have revised the section to read:

"A detailed chronological frame for the Li/Mg records was established with the aid of the generalized annual record of remote sensing SST data (JPL MUR, daily averaged 2003 – 2018) (Fig. 2). This data demonstrates average calendar dates of seasonal extremes to occur on 31 May during spring intermonsoon (SIM), on 15 August during southwest monsoon (SWM), on 25 October during autumn intermonsoon (AIM) and on 06 February during northeast monsoon (NEM). Average dates of inflection points between consecutive seasons from the generalized annual SST record are 18 March (NEM – SIM), 25 June (SIM – SWM), 30 September (SWM – AIM) and 10 December (AIM – NEM)."

- **3.17.** "Note on the standard corals: How do we know that the other corals are effective standards for comparison? Who is to say that they were not influenced by SST or OA? Or other factors?"
- AR: We do not exclude the influence of environmental factors other than SST on the calcification of the standard corals (GBR and Hawaii) at any point. Based on the highly convincing correlations with SST (bulk density:  $r^2 = 0.49$ , p < 0.0001; extension rate:  $r^2 = 0.9$ , p < 0.0001), we stated that calcification of these corals is "largely controlled" by water temperature (Lough and Barnes, 2000).

**Results**

- 3.18. "Lines 248-249: Do these n refer to the number of transects for the measures?"
- AR: These n refer to the number of years that were used to calculate the mean annual bulk density of the individual coral samples.

**Discussion**

- **3.19.** "A lot of the current discussion is results. These should be moved to the results section and then the discussion can include more incorporation of implications/meaning of these results."
- AR: Thank you very much for this advice. We have moved several parts of the discussion (e.g., an updated version of Figure 9) to the results.
- 3.20. "Lines 285-288: But don't you expect biological variability?"
- AR: This quantitative comparison of our data with data of *Porites* from the Indo-Pacific no longer appears in the revised version of our manuscript (see RC2-3.02).
- 3.21. "Lines 367-369: Split into two different sentences for easier reading."
- AR: We have revised the section to read: "This finding implies that there is no intensified upregulation of internal  $\Omega_{cf}$  relative to  $\Omega_{sw}$  during the non-upwelling seasons (McCulloch et al., 2017; DeCarlo et al., 2018; D'Olivo and McCulloch, 2017). As an explanation, we propose that internal upregulation processes of corals affected by seasonal upwelling are not capable to adapt completely to ocean chemistry change on a quarterly scale. As a consequence, a relatively low  $\Omega_{cf}$  is maintained year-round so as to avoid high gradients to the external  $\Omega_{sw}$  during southwest monsoonal upwelling."
- **3.22.** *"Lines 414-433: This section is lacking incorporation of the current literature and needs some more grounding in terms of what is known and previous work."*

AR: The implication of our results described in this section largely refers to the enhanced annular mean extension rates and the lower skeletal density observed in *Porites* from Masirah Island compared to the reference data from the Indo-Pacific (Lough and Barnes, 2000). Given the reframing of the discussion part, this section will undergo a major reorganisation. The revised focus will then be on the implications derived from the comparison of intra-annual patterns of calcification (upwelling season vs. non-upwelling season) (see chapter 4.3. of the discussion paper: "Monthly records of coral calcification"). A revised key aspect will be the finding of a constant and year-round relatively low skeletal density regardless of the mere 3-month time span of upwelling (disscussed in lines 364-376 in the discussion paper).

**Figures**

- **3.23.** "Figure 1: This is a really helpful figure to demonstrate this reef system, collection site, and the currents/upwelling, however, this figure could be made a bit clearer with a few updates as suggested here. There is a lot going on with colours so I recommend making your land either white or grey to make the focus of the map more on the reef locations. I also recommend selecting a different colour to represent the coral reef provinces with better contrast against the red and blue."
- AR: We have turned the colour of the land surface greyish to put more focus on the reef locations. The yellow colour of the coral reef provinces contrasts most strongly with the adjacent red and blue colours, which is why we consider it to be exceptionally suitable.
- 3.24. "Figure 2/3: Please define NEM, SIM, SWM, and AIM in your figure captions."
- AR: We have defined the abbreviations in the figure captions.
- 3.25. "Figure 3: please include what years were assessed in to calculate these monthly values."
- AR: We have included the number of years used for the calculation of the multi-year monthly means.
- 3.26. "Figure 5: again, please include the years assessed in the monthly values."
- AR: We have included the number of years used for the calculation of the multi-year monthly means.

**4. References**

[revised manuscript text omitted]

---

## Author Comment (AC3)

Authors' response to reviews of

**"Nutrients attenuate the negative effect of ocean acidification on reef coral calcification in the Arabian Sea upwelling zone (Masirah Island, Oman)"**

Philipp M. Spreter et al.

*Biogeosciences*, doi.org/10.5194/bg-2021-213-RC2, 2021

RC: Reviewers' comment      AR: Authors' response

**1. Introduction:**

on behalf of my co-authors, I would like to thank the reviewers for commenting on our manuscript. Their supporting comments and ideas are important for us to improve the significance of our work.

Given that both reviewers stated fundamental concerns about the framing and interpretation of our data, I would like to address these general aspects in advance, before discussing the single reviewer comments in detail.

The reviewers express general doubts about the reliability of the mean annual calcification data (extension rate and bulk density) of *Porites* from Masirah Island reported by us because of a comparatively small number of replicates (n=3). Accordingly, low mean annular skeletal bulk density and enhanced linear extension rate of *Porites* from Masirah Island recorded by our study relative to those of *Porites* from the Indo-Pacific is suggested to be a possible artefact of a small database. However, a larger number of specimens and respective calcification data is available to us. This supplementary information was provided to the open discussion in a reply to RC1 (16 Sept 2021). The dataset clearly shows that the calcification data presented in our discussion paper are not biased by poor statistics, but document a mean situation for the region. We selected three out of the six samples for further geochemical analysis (laser ICP-MS) that were considered representative for the site studied. These geochemical analyses enabled us to obtain monthly resolved calcification patterns of *Porites* from a region affected by intense seasonal upwelling, which is indeed a novel approach. We will therefore keep the focus on these three samples also in the revised version of the manuscript. However, as no *Porites* growth data exists from the Arabian Sea, the entire data of all 6 specimen will be added to the revision as an electronic supplement. In this way, we hope to encourage further *Porites* growth studies in a hitherto unexplored region of the western Indian Ocean.

Being aware that the number of three samples is statistically problematic, we will re-arrange the manuscript as suggested by both reviewers and place the focus more on a combination of the proxy data (e.g. Li/Mg, Ba/Ca) and intra-annual growth patterns presented in the original manuscript. To investigate the influence of upwelling on coral calcification, we discuss differences between the seasonal growth patterns within our samples, in particular those observed during the upwelling and non-upwelling seasons. By doing so, "intra-reef variabilities" in coral growth problematized by Reviewer 2 are largely negligible, as local factors and the genotype remain constant within a single specimen. We strongly feel discussing the existing data in this way is scientifically more robust and solves any legitimate concerns of the reviewers.

**2. Reviewer # 1 (RC1):**

**2.1.**      *"Three replicate cores is just not enough to have robust statistics…Actually, the only way I see forward for the authors (besides getting more corals) is to completely re-frame the study based on the geochemistry."*

AR:      This comment is addressed in detail in the introduction.

**2.2.**      *"I had no idea there was going to be Li/Mg geochemistry in this study based on the title and abstract."*

AR:      Information on the application of Li/Mg and Ba/Ca trace element analyses for the interpretation of intra-annual patterns of coral calcification will be provided in the revised version of the abstract.

**2.3.**      *"I don't think it makes sense to compare the growth rates to just the GBR."*

AR:      In the discussion paper, we compare our growth data from Masirah Island with reference data from the GBR (29 sites, 245 colonies) and from Hawaii (14 sites, 140 colonies) (Grigg, 1981; Lough and Barnes,

2000). These two datasets represent the largest of their kind for *Porites*. The calcification variables (skeletal density, extension rate and calcification rate) are all significantly correlated with SST (Lough and Barnes, 2000). For this reason, the regressions of these relationships are frequently used as "baselines" for estimating the performance of recent and fossil *Porites* with regard to their bulk density, extension rates and calcification rate at a certain SST (Brachert et al., 2016; Brachert et al., 2020; Lough et al., 2016; Manzello et al., 2014). However, we agree that our approach of applying the regressions from the Indo-Pacific reference dataset to predict calcification performance of *Porites* from the Arabian Sea possibly overstresses the global validity of the causal relationship between SST and calcification because of a small sample number of specimens in our dataset (n = 3). For this reason, in our revised manuscript, we refrain from predicting "theoretical values" for the calcification variables from a given SST and instead provide a more qualitative comparison of our data with published *Porites* growth data. In this way, we will provide a more global perspective, by showing additional (published) *Porites* growth data from Thailand, Western Australia, the Gulf of Mexico plus three other regions affected by upwelling, in addition to the previously presented data from the GBR and Hawaii. All published records from globally distributed upwelling sites show a conspicuous and consistent pattern of low skeletal density without any exception (Fig.1).

[Figure]

**Figure 1:** Annual means in *Porites* skeletal bulk density versus annual sea surface temperature (SST) from various Indo-Pacific and Western Atlantic reef sites. Note that all reef sites affected by seasonal upwelling show exceptionally low skeletal bulk density.

**3. Reviewer # 2 (RC2):**

**Concerns:**

**3.01.** *"Overall, an n of 3 is fairly small to make these growth assumptions with… growth is highly variable across locations and genotypes and thus would require a higher n to make stronger predictions/estimates."*

AR: This comment is addressed in detail in the introduction.

**3.02.** *"Coral growth is reliant on a lot of different biological processes that are largely ignored throughout this text. While calcifying fluid is discussed, the authors never actually make these measurements. Further, the authors ignore how much variability in coral growth there is within a genus or even a species and over-estimate the reliance of growth on SST. The authors should consider what other factors can impact the variability across their cores and work to put it more in the context of the biology of corals. Along these same lines, coral cores collected from the Pacific may not experience similar environmental conditions and may likely have different populations/species of Porites corals, thus comparison of the corals from this study with Pacific corals should be made with caution."*

AR: This reviewer comment will be split into three sections, which will be answered separately in the following:

*"Coral growth is reliant on a lot of different biological processes that are largely ignored throughout this text."*

AR: We are working on dead coral material and do not want to speculate about the biological processes in detail, but rather discuss coral calcification with respect to environmental controls. In this context, we strongly feel to address fundamental environmental factors such as SST, pH, $\Omega_{Arag}$ and nutrients which have been shown to have the greatest impact on reef coral calcification (Courtney et al., 2017; Ross et al., 2019).

„*While calcifying fluid is discussed, the authors never actually make these measurements."*

AR: Indeed, we did not perform combined analyses of B/Ca and $\delta^{11}B$ on our specimens in order to estimate the $\Omega_{cf}$ of the calcification fluid. Recent publications, however, have led us to consider $\Omega_{cf}$ as a major parameter controlling skeletal density (Mollica et al., 2018). Considering this, it is essential for us to discuss the variability in skeletal density with regard to possible influences of $\Omega_{cf}$. Systematic measurements of B/Ca and $\delta^{11}B$ are currently underway in the frame of a cooperation beyond the group of co-authors for this manuscript.

*"Further, the authors ignore how much variability in coral growth there is within a genus or even a species and over-estimate the reliance of growth on SST. Along these same lines, coral cores collected from the Pacific may not experience similar environmental conditions and may likely have different populations/species of Porites corals, thus comparison of the corals from this study with Pacific corals should be made with caution"*

AR: An overwhelming number of studies demonstrate SST to have the most significant impact on reef coral calcification (Courtney et al., 2017; Lough and Barnes, 2000; Lough, 2008; Lough et al., 2016). For this reason, we do not agree with the statement of having overestimated the reliance of growth on SST. However, as we already mentioned in our response to RC1-2.3, we agree that our approach of applying the regressions from the Indo-Pacific reference dataset to predict calcification performance of *Porites* from the Arabian Sea likely poses problems. This is because of a small number of replicates (n = 3), which means that we are likely unable to capture possible full growth variability at Masirah Island. To overcome this, we decided to refrain from calculating theoretical calcification based on the regressions from Indo-Pacific *Porites* but rather to focus on the intra-annual growth variability between the upwelling season and the non-upwelling season at Masirah Island. Intra-annual growth variability within a single specimen is now being discussed in terms of the coral's calcification response to seasonally changing environmental factors such as SST, $\Omega_{Arag}$, light availability for photosynthesis and seawater nutrients, while growth variability across our samples is also discussed with regard to locational- and inter-species variability.

**3.03.** *"While the satellite SST matched very well with in situ SST measurements, the other satellite phosphate and nitrate parameters were not similarly grounded in truth. Further, the lower resolution for these satellite products is concerning since nutrient values can be highly variable across a spatial gradient."*

AR: WOA18 nutrient data is not a satellite product, but a generalized interpolation of all available in-situ observations performed within each 1° square (Garcia et al., 2018). Due to reframing of our study with a new focus on intra-annual growth patterns, we only consider monthly resolved WOA18 nutrient data from Masirah Island in the revised version of the manuscript (Fig.2, discussion paper). These data show relatively homogeneous nutrient concentrations during the non-upwelling season and a pronounced maximum of both essential nutrients (phosphate, nitrate) during the upwelling season. This intra-annual dynamic of nutrient concentrations is generally associated with upwelling areas and has also been observed in the northern Arabian Sea through in-situ observations (Currie et al., 1973; Savidge et al., 1990). In combination with the excellent agreement to skeletal Ba/Ca records, we assume that the general pattern of monthly resolved WOA18 nutrient concentrations at Masirah Island qualitatively reflect the intra-annual nutrient cycle of surface waters (Lea et al., 1989; Montaggioni et al., 2005; Tudhope et al., 1996). Nevertheless, we are aware of temporal and spatial heterogeneities in nutrient distribution and discuss this based on variabilities in the Ba/Ca records across specimen.

**3.04.** *"Finally, because of the discussion of ocean acidification is a major component of this manuscript, it also may be valuable for the authors to also include carbonate parameters for the area (i.e., omega-aragonite, TA, pH, etc.)."*

AR:     We regret that except for data from the Arabian Sea Expedition (1995) recorded during the southwest monsoon (upwelling) and northeast monsoon (non-upwelling), no additional information on the carbonate chemistry of the northern Arabian Sea is available. Existing data from the Arabian Sea Expedition on seawater aragonite saturation ($\Omega_{sw}$) are reported in the methods section of the discussion paper (lines 101-105) and are discussed in the following (line 370).

**Minor edits:**

**Abstract**

***3.05.***     *"Lines 14-15: This statement is a bit of an overstatement. Calcification responses to changing environments is very well studied. If this is intended to be in terms of a specific type of environmental variation, then that needs to be clarified."*

AR:     We have revised the sentence to read: "The calcification response of reef corals to rapid and simultaneous changes in $\Omega_{sw}$ and seawater nutrient concentrations is not well studied, however."

**Introduction**

***3.06.***     *"Line 29: Consider replacing 'zooxanthellate' with symbiotic."*

AR:     We have replaced the word "zooxanthellate" by "symbiotic".

***3.07.***     *"Lines 36-37: This statement should be backed with the literature. Coral calcification is not highly debated; however, responses are highly variable. I recommend referencing several papers covering this here."*

AR:     We have revised the sentence to read: "The responses of reef coral calcification on this rapidly changing environment are highly variable and remain currently a matter of debate." and covered this statement with literature from Cornwall et al., (2021), Guan et al., (2020) and Hall et al., (2018).

***3.08.***     *"Lines 50-52: This statement seems misplaced and should be incorporated better within the introduction. Additionally, recent reviews suggest different calcification responses in corals under global change (see Cornwall et al 2021, Global declines in coral reef calcium carbonate production under ocean acidification and warming, PNAS)."*

AR:     We have removed this sentence because variable responses in corals under global change are already addressed in the introduction where the reference to the work of Cornwall et al., (2021) is cited (see RC2-3.07).

***3.09.***     *"Line 54: savage disposal? Do you mean sewage?"*

AR:     We have revised the spelling mistake to "sewage".

***3.10.***     *"Lines 54-55: This is an incomplete statement."*

AR:     We have revised the sentence to read: "Eutrophication can have both beneficial as well as detrimental effects on coral growth, however (Tomascik and Sander, 1985; Tomascik, 1990)."

***3.11.***     *"Lines 55-56: Again, I suggest updating your language here to reflect more recent terminology of coral algal symbionts (see LaJeunesse et al. 2018, Systematic Revision of Symbiodiniaceae Highlights the Antiquity and Diversity of Coral Endosymbionts, Current Biology)."*

AR:    We have revised the sentence to read: "Reef corals are highly adapted to oligotrophic waters with micro-algae symbionts to allow an efficient use of essential nutrients and to outcompete other fast-growing biota on a reef whose growth is inhibited by the undersupply of nutrients (Barrot and Rohwer, 2012; LaJeunesse et al., 2018; Vermeij et al., 2010)."

*3.12.*   *"Lines 66-68: These sentences could be a bit stronger to introduce this important topic in your introduction."*

AR:    We have revised the sentence to read: "Understanding how coral calcification responds to rapid changes in seawater nutrient conditions and $\Omega_{sw}$ is critical for more accurate predictions on the persistence of reef habitats under the influence of global change."

*3.13.*   *"Lines 70-72: This statement would benefit from a clear connection of how calcification responses from upwelling locations can be applied to systems without upwelling."*

AR:    We have revised the sentence to read: "This allows these regions to serve as natural laboratories to investigate the calcification response of reef corals to these multiple environmental stressors that are likely to affect global coral reefs in the near future (Camp et al., 2018; Wizemann et al., 2018)."

*3.14.*   *"Lines 84-125: This section should be moved into the methods as a section describing the sites. A condensed version of this could be included in the previous paragraph to describe the sites if that is desired."*

AR:    We have moved the chapter "Arabian Sea climate and oceanography" to the beginning of the methodology section in the revised version of the manuscript.

**Methods**

*3.15.*   *"Lines 128-134: It sounds like these colonies were not collected in situ, rather collected as dead skeletons from the beach. What about differences in local conditions? If these corals were washed up on shore you don't really know what depth or location they came from? Also what years? How do you know when they washed up on the beach?"*

AR:    This reviewer comment will be split into three sections, which will be answered separately in the following:

       *"…when were they washed up on the beach?"*

AR:    Certainly, all the corals were washed up on the beach by the same event, as they were all collected from the same storm deposit, which is attributed to Cyclone Gonu in 2007 (Fritz et al., 2010).

       *"…also what years?"*

AR:    The detailed information about the calendar years represented in our records is marginal, as we establish generalized annual calcification records with monthly resolution from several years of the total record lengths (Multi-year monthly means). These generalised records represent the mean monthly calcification of *Porites* corals at Masirah Island during the more recent current era.

       *"…What about differences in local conditions…depth or location?"*

AR:    In the revised version of our manuscript, we discuss intra-annual variability of calcification (in particular between the upwelling season and non-upwelling season) with regard to seasonal environmental changes. Location factors are hereby to be neglected, as they remain uniform within a single specimen.

*3.16.*   *"Lines 216-219: Unclear what these numbers and acronyms represent. Please rephrase in a clearer way."*

AR:     These acronyms refer to the two monsoon seasons (SWM = Southwest Monsoon, NEM = Northeast Monsoon) and the intermonsoon seasons (SIM = Spring Intermonsoon, AIM = Autumn Intermonsoon), respectively, and are introduced in the preceding chapter "Arabian Sea climate and oceanography" (lines 88-92) of the discussion paper. However, to ensure better reading in the section "Data matching and age model development" (lines 216-216), we have revised the section to read:

"A detailed chronological frame for the Li/Mg records was established with the aid of the generalized annual record of remote sensing SST data (JPL MUR, daily averaged 2003 – 2018) (Fig. 2). This data demonstrates average calendar dates of seasonal extremes to occur on 31 May during spring intermonsoon (SIM), on 15 August during southwest monsoon (SWM), on 25 October during autumn intermonsoon (AIM) and on 06 February during northeast monsoon (NEM). Average dates of inflection points between consecutive seasons from the generalized annual SST record are 18 March (NEM – SIM), 25 June (SIM – SWM), 30 September (SWM – AIM) and 10 December (AIM – NEM)."

**3.17.**  *"Note on the standard corals: How do we know that the other corals are effective standards for comparison? Who is to say that they were not influenced by SST or OA? Or other factors?"*

AR:     We do not exclude the influence of environmental factors other than SST on the calcification of the standard corals (GBR and Hawaii) at any point. Based on the highly convincing correlations with SST (bulk density: $r^2 = 0.49$, $p < 0.0001$ ; extension rate: $r^2 = 0.9$, $p < 0.0001$), we stated that calcification of these corals is "largely controlled" by water temperature (Lough and Barnes, 2000).

**Results**

**3.18.**  *"Lines 248-249: Do these n refer to the number of transects for the measures?"*

AR:     These n refer to the number of years that were used to calculate the mean annual bulk density of the individual coral samples.

**Discussion**

**3.19.**  *"A lot of the current discussion is results. These should be moved to the results section and then the discussion can include more incorporation of implications/meaning of these results."*

AR:     Thank you very much for this advice. We have moved several parts of the discussion (e.g., an updated version of Figure 9) to the results.

**3.20.**  *"Lines 285-288: But don't you expect biological variability?"*

AR:     This quantitative comparison of our data with data of *Porites* from the Indo-Pacific no longer appears in the revised version of our manuscript (see RC2-3.02).

**3.21.**  *"Lines 367-369: Split into two different sentences for easier reading."*

AR:     We have revised the section to read:  "This finding implies that there is no intensified upregulation of internal $\Omega_{cf}$ relative to $\Omega_{sw}$ during the non-upwelling seasons (McCulloch et al., 2017; DeCarlo et al., 2018; D'Olivo and McCulloch, 2017). As an explanation, we propose that internal upregulation processes of corals affected by seasonal upwelling are not capable to adapt completely to ocean chemistry change on a quarterly scale. As a consequence, a relatively low $\Omega_{cf}$ is maintained year-round so as to avoid high gradients to the external $\Omega_{sw}$ during southwest monsoonal upwelling."

**3.22.**  *"Lines 414-433: This section is lacking incorporation of the current literature and needs some more grounding in terms of what is known and previous work."*

AR:     The implication of our results described in this section largely refers to the enhanced annular mean extension rates and the lower skeletal density observed in *Porites* from Masirah Island compared to the reference data from the Indo-Pacific (Lough and Barnes, 2000). Given the reframing of the discussion part, this section will undergo a major reorganisation. The revised focus will then be on the implications derived from the comparison of intra-annual patterns of calcification (upwelling season vs. non-upwelling season) (see chapter 4.3. of the discussion paper: "Monthly records of coral calcification"). A revised key aspect will be the finding of a constant and year-round relatively low skeletal density regardless of the mere 3-month time span of upwelling (disscussed in lines 364-376 in the discussion paper).

**Figures**

***3.23.***    *"Figure 1: This is a really helpful figure to demonstrate this reef system, collection site, and the currents/upwelling, however, this figure could be made a bit clearer with a few updates as suggested here. There is a lot going on with colours so I recommend making your land either white or grey to make the focus of the map more on the reef locations. I also recommend selecting a different colour to represent the coral reef provinces with better contrast against the red and blue."*

AR:     We have turned the colour of the land surface greyish to put more focus on the reef locations. The yellow colour of the coral reef provinces contrasts most strongly with the adjacent red and blue colours, which is why we consider it to be exceptionally suitable.

***3.24.***    *"Figure 2/3: Please define NEM, SIM, SWM, and AIM in your figure captions."*

AR:     We have defined the abbreviations in the figure captions.

***3.25.***    *"Figure 3: please include what years were assessed in to calculate these monthly values."*

AR:     We have included the number of years used for the calculation of the multi-year monthly means.

***3.26.***    *"Figure 5: again, please include the years assessed in the monthly values."*

AR:     We have included the number of years used for the calculation of the multi-year monthly means.

**4. References**

Barrot, K.T., Rohwer, F.L.: Unseen players shape benthic competition on coral reefs. Trends Microbiol., 20, 621-628, http://doi.org/10.1016/j.tim.2012.08.004, 2012.

Camp, E.F., Schoepf, V., Mumby, P.J., Hardtke, L.A., Rodolfo-Metalpa, R., Smith, D.J., Suggett, D.J.: The future of coral reefs subject to rapid climate change: Lessons from natural extreme environments. Front. Mar. Sci., 5, 1–21, https://doi.org/10.3389/fmars.2018.00004, 2018.

Cornwall, C.E., Comeau, S., Kornder, N.A., Perry, C.T., van Hooidonk, R., DeCarlo, T.M., Pratchett, M.S., Anderson, K.D., Browne, N., Carpenter, R., Diaz-Pulido, G., D'Olivo, J.P., Doo, S.S., Figueiredo, J., Fortunato, S.A.V., Kennedy, E., Lantz, C.A., McCulloch, M.T., González-Rivero, M., Schoepf, V., Smithers, S.G., Lowe, R.J.: Global declines in coral reef calcium carbonate production under ocean acidification and warming. Proc. Natl. Acad. Sci. U. S. A., 118, 1-10, https://doi.org/10.1073/pnas.2015265118, 2021.

Currie, R.I., Fisher A.E., Hargreaves P.M.: Arabian Sea Upwelling. In: Zeitzschel B., Gerlach S.A., The Biology of the Indian Ocean. Ecological Studies (Analysis and Synthesis) 3. Springer, Berlin, Heidelberg, http://doi.org/10.1007/978-3-642-65468-8, 1973.

Courtney, T.A., Lebrato, M., Bates, N.R., Collins, A., De Putron, S.J., Garley, R., Johnson, R., Molinero, J.C., Noyes, T.J., Sabine, C.L., Andersson, A.J.: Environmental controls on modern scleractinian coral and reef-scale calcification. Sci. Adv. 3, 1-9, https://doi.org/10.1126/sciadv.1701356, 2017.

Elizalde-Rendón, E.M., Horta-Puga, G., Gonzáles-Dias, P., Carricart-Ganivet, J.P.: Growth characteristics of the reef-building coral Porites asteroids under different environmental conditions in the Western Atlantic. Coral Reefs, 29, 607-614, https://doi.org/10.1007/s00338-010-0604-7, 2010.

Garcia, H. E., Weathers, K., Paver, C. R., Smolyar, I., Boyer, T. P., Locarnini, R. A., Zweng, M. M., Mishonov, A. V., Baranova, O. K., Reagan, J. R.: World Ocean Atlas 2018, 4: Dissolved Inorganic Nutrients (phosphate, nitrate, silicate), 2019.

Guan, Y., Hohn, S., Wild, C., Merico, A.: Vulnerability of global coral reef habitat suitability to ocean warming, acidification and eutrophication. Glob. Chang. Biol., 26, 5646–5660, https://doi.org/10.1111/gcb.15293, 2020.

Grigg, W.G.: Coral reef development at high latitudes in Hawaii. Proceedings of the Fourth International Coral Reef Symposium, Manila, 1, 688-693, 1981.

Hall, E.R., Muller, E.M., Goulet, T., Bellworthy, J., Ritchie, K.B., Fine, M.: Eutrophication may compromise the resilience of the Red Sea coral Stylophora pistillata to global change. Mar. Pollut. Bull., 131, 701–711, https://doi.org/10.1016/j.marpolbul.2018.04.067, 2018.

LaJeunesse, T.C., Parkinson, J.E., Gabrielson, P.W., Jeong, H.J., Reimer, J.D., Voolstra, C.R., Santos, S.R.: Systematic Revision of Symbiodiniaceae Highlights the Antiquity and Diversity of Coral Endosymbionts. Curr. Biol., 28, 2570-2580, https://doi.org/10.1016/j.cub.2018.07.008, 2018.

Lea, D.W., Shen, G.T., Boyle, E.A.: Coralline barium records temporal variability in equatorial pacific upwelling. Nature, 340, 373-376, https://doi.org/10.1038/340373a0, 1989.

Lough, J.M., Barnes, D.J.: Environmental controls on growth of the massive coral Porites. J. Exp. Mar. Bio. Ecol., 245, 225-243, http://doi.org/10.1016/s0022-0981(99)00168-9, 2000.

Montaggioni, L.F., Le Cornec, F., Corrège, T., Cabioch, G.: Coral barium/calcium record of mid-Holocene upwelling activity in New Caledonia, South-West Pacific. Palaeogeogr., Palaeoclimatol., Palaeoecol., 273, 436-455, https://doi.org/10.1016/j.palaeo.2005.12.018, 2005.

Ross, C.L., DeCarlo, T.M., McCulloch, M.T.: Environmental and physiochemical controls on coral calcification along a latitudinal temperature gradient in Western Australia. Glob. Chang. Biol., 25, 431–447, https://doi.org/10.1111/gcb.14488, 2019.

Savidge, G., Lennon, J., Matthews, A.J.: A shore-based survey of upwelling along the coast of Dhofar region, southern Oman. Continental Shelf Research, https://doi.org/ 10.1016/0278-4343(90)90022-E, 259-275, 1990.

Scoffin, T.P., A.W. Tudhpe, B.E. Brown, H.C. and R.F.C.: Pattern and possible environmental controls of skeletogenesis of Porites lutea, South Thailand. Coral Reefs, 11, 1–11, https://doi.org/10.1007/BF00291929, 1992.

Tomascik, T., Sander, F.: Effects of eutrophication on reef-building corals. Mar. Biol., 87, 143–155, https://doi.org/10.1016/0198-0254(87)90298-6, 1985.

Tomascik, T.: Growth rates of two morphotypes of Montastrea annularis along a eutrophication gradient, Barbados, W.I. Mar. Pollut. Bull., 21, 376–381, https://doi.org/10.1016/0025-326X(90)90645-O, 1990.

Tudhope, A.W., Lea, D.W., Shimmield, G.B., Chilcott, C.P., Head, S.: Monsoon climate and Arabian Sea coastal upwelling recorded in massive corals from southern Oman. Palaios, 11, 347-361, https://doi.org/10.2307/3515245, 1996.

Vermeij, M.J.A., van Moorselaar, I., Engelhard, S., Hörnlein, C., Vonk, S.M., Visser, P.M.: The effect of nutrient enrichment and herbivore abundance on the ability of turf algae to overgrow coral in the Caribbean. PLoS One, 5, 1-8, https://doi.org/10.1371/journal.pone.0014312, 2010.

Wizemann, A., Nandini, S.D., Stuhldreier, I., Sánchez-Noguera, C., Wisshak, M., Westphal, H., Rixen, T., Wild, C., Reymond, C.E.: Rapid bioerosion in a tropical upwelling coral reef. PLoS One, 13, 1–22, https://doi.org/10.1371/journal.pone.0202887, 2018.

---

## Author Response (AR1)

Point-by-point description of changes made to the revised manuscript

**"Nutrients attenuate the negative effect of ocean acidification on reef coral calcification in the Arabian Sea upwelling zone (Masirah Island, Oman)"**

Philipp M. Spreter et al.

*Biogeosciences*, doi.org/10.5194/bg-2021-213, 2021

We would like to thank the reviewers for commenting on our manuscript. Their supporting comments and ideas have substantially improved the scientific significance of our work. In line with the reviewers' comments and the editor recommendations, major changes have been made to our revised manuscript. General modifications include:

- Improved statistical robustness of the mean annual calcification data through addition of three further specimens (Table 2, Fig.9).

- Discussion of the sub-annual growth patterns is made internally, i.e., by comparing the calcification performance during the upwelling and non-upwelling season (line 321-362).

- More extensive discussion of the geochemical data (289-312).

- Reorganisation of paragraphs into other chapters for reaching a clear separation between results and discussion as well as between introduction and methods.

- Omission of a quantitative comparison of our data with growth data from the Indo-Pacific. In the revised manuscript, this approach has been replaced by a qualitative comparison of our data with an extended data set on *Porites* calcification from various Pacific and Atlantic reef sites (Fig. 9).

- Adjustment of the Abstract and the Conclusion.

- Change of the working title to:

   **„Calcification response of reef corals to seasonal upwelling in the
   northern Arabian Sea (Masirah Island, Oman)"**

Detailed responses to all comments made by the reviewers and the editor are described point-by-point in the section below.

**1. Editor comments:**

*1.1.    ...”the manuscript should be refocused on the geochemical proxy data instead of the calcification data”*

The geochemical proxy data is discussed more extensively in the revised manuscript (line 289-312). However, geochemistry remains a tool for establishing sub-annual chronologies of the calcification records and for assessing the influence of temperature on skeletal calcification only. The *Porites* calcification data represents the very first contribution to this topic for the Arabian Sea and the north-western Indian Ocean. For this reason, we are of the opinion that the focus of the manuscript should be kept on the calcification data.

*1.2.    ...”you rely heavily on low average saturation states in this region, however, there is minimal data to back this up in any of your regressions and plots”*

To back up the presence of a temporarily low seawater aragonite saturation state ($\Omega_{sw}$) at the sampling site, we included monthly-modelled $\Omega_{sw}$ to Fig.2 (Takahashi et al., 2014).

**2. Reviewer # 1 (RC1):**

*2.1.* *"Three replicate cores is just not enough to have robust statistics…Actually, the only way I see forward for the authors (besides getting more corals) is to completely re-frame the study based on the geochemistry."*

Statistical robustness of the mean annual calcification data has been reached through addition of three further specimens (Table 2; Fig.9). A total of six specimens for estimating mean calcification performance at a site is in line with other coral growth studies from the literature (e.g., Lough and Barnes, 2000; Manzello et al., 2014).

*2.2.* *"I had no idea there was going to be Li/Mg geochemistry in this study based on the title and abstract."*

We have added information on the application of Li/Mg and Ba/Ca geochemistry to the abstract (line 18).

*2.3.* *"I don't think it makes sense to compare the growth rates to just the GBR."*

We fully refrain from the quantitative comparison of our data with growth data from the Indo-Pacific (GBR and Hawaii). Instead, we show a more qualitative comparison of our data with published *Porites* growth data (Fig.9). In this way, we provide a more global perspective, by showing published *Porites* growth data from Thailand, Western Australia, the Gulf of Mexico plus three other regions affected by upwelling, in addition to the previously presented data from the GBR and Hawaii. This approach fully supports our finding of low skeletal density and high extension growth of Porites from upwelling zones.

**3. Reviewer # 2 (RC2):**

**Concerns:**

*3.1.* *"Overall, an n of 3 is fairly small to make these growth assumptions with… growth is highly variable across locations and genotypes and thus would require a higher n to make stronger predictions/*estimates.*"*

The statistical robustness of the mean annual calcification data is now sufficient through addition of three further specimens (Table 2). The total sample size of six corals for a site is in line with other coral growth studies from the literature (e.g., Lough and Barnes, 2000; Manzello et al., 2014).

*3.2.* *"Coral growth is reliant on a lot of different biological processes that are largely ignored throughout this text."*

In the revised manuscript, we describe in detail that photosynthesis by algal symbionts provides most of the energy for the coral to grow (line 40-45, 341-344). This highlights the variety of biological processes controlling photosynthetic efficiency to also affect coral growth. Furthermore, we introduce and discuss the process of active pH and $\Omega_{Arag}$ upregulation within the corals' calcification fluid (line 47-52, 358-362, 391-395). We therefore do believe to have adequately addressed major controlling factors on coral calcification, i.e. SST, light, nutrients, pH and $\Omega_{Arag}$ (Courtney et al., 2017; Ross et al., 2019).

*3.3.* *„While calcifying fluid is discussed, the authors never actually make these measurements."*

Indeed, we did not perform combined analyses of B/Ca and $\delta^{11}B$ on our specimens in order to estimate the $\Omega_{cf}$ of the calcification fluid. Recent publications, however, have led us to consider $\Omega_{cf}$ as a major parameter controlling skeletal density (Mollica et al., 2018). Considering this, it is essential for us to discuss the variability in skeletal density with regard to possible influences of $\Omega_{cf}$. Systematic measurements of B/Ca and $\delta^{11}B$ are currently underway in the frame of a cooperation beyond the group of co-authors for this manuscript.

3.4. *"Further, the authors ignore how much variability in coral growth there is within a genus or even a species and over-estimate the reliance of growth on SST. Along these same lines, coral cores collected from the Pacific may not experience similar environmental conditions and may likely have different populations/species of Porites corals, thus comparison of the corals from this study with Pacific corals should be made with caution"*

In the revised version of the manuscript, we completely refrain from comparing our data with "theoretical values" based on temperature-calcification calibrations of *Porites* from the Indo-Pacific. Nonetheless, it should be noted that studies on Pacific *Porites* did explicitly make no difference between species (Lough et al., 2000)

3.5. *"While the satellite SST matched very well with in situ SST measurements, the other satellite phosphate and nitrate parameters were not similarly grounded in truth. Further, the lower resolution for these satellite products is concerning since nutrient values can be highly variable across a spatial gradient."*

In the revised version of the manuscript we only present monthly resolved seawater nutrient data from Masirah Island (WOA18, Garcia et al., 2018) (Fig.2). Considering the excellent agreement to skeletal Ba/Ca records (line 209-211), we assume that the general pattern of monthly resolved WOA18 nutrient concentrations at the study site qualitatively reflect the intra-annual nutrient cycle of surface waters (Montaggioni et al., 2006). We are aware of temporal and spatial heterogeneities in nutrient distribution and discuss this based on variability between the Ba/Ca records across specimen (line 293-294).

3.6. *"Finally, because of the discussion of ocean acidification is a major component of this manuscript, it also may be valuable for the authors to also include carbonate parameters for the area (i.e., omega-aragonite, TA, pH, etc.)."*

In addition to the reference from Omer (2010) which gives information on the seawater $\Omega_{Arag}$ at Masirah Island in lines 98-100, we have added monthly-modelled data on seawater $\Omega_{Arag}$ in Fig.2 (Takahashi et al., 2014).

**Minor edits:**

**Abstract**

3.7. *"Lines 14-15: This statement is a bit of an overstatement. Calcification responses to changing environments is very well studied. If this is intended to be in terms of a specific type of environmental variation, then that needs to be clarified."*

We have revised the sentence to read: "The calcification response of reef corals on rapid changes in $\Omega_{sw}$ and seawater nutrient concentrations is currently under discussion in coral science."

**Introduction**

3.8. *"Line 29: Consider replacing 'zooxanthellate' with symbiotic."*

We have replaced the word "zooxanthellate" by "symbiotic".

3.9. *"Lines 36-37: This statement should be backed with the literature. Coral calcification is not highly debated; however, responses are highly variable. I recommend referencing several papers covering this here."*

We have revised the sentence to read: "The responses of reef coral calcification on this rapidly changing environment are highly variable and remain contemporarily a matter of intense research." and covered this statement with literature from Cornwall et al., (2021), Guan et al., (2020) and Hall et al., (2018).

3.10. *"Lines 50-52: This statement seems misplaced and should be incorporated better within the introduction. Additionally, recent reviews suggest different calcification responses in corals under global change (see Cornwall et al 2021, Global declines in coral reef calcium carbonate production under ocean acidification and warming, PNAS)."*

We have removed this sentence because variable responses in corals under global change are already addressed in the introduction where the reference to the work of Cornwall et al., (2021) is cited.

3.11. *"Line 54: savage disposal? Do you mean sewage?"*

We have revised the spelling mistake to "sewage".

3.12. *"Lines 54-55: This is an incomplete statement."*

We have revised the sentence to read: "Eutrophication can have both beneficial as well as detrimental effects on coral growth, however (Tomascik and Sander, 1985; Tomascik, 1990)."

3.13. *"Lines 55-56: Again, I suggest updating your language here to reflect more recent terminology of coral algal symbionts (see LaJeunesse et al. 2018, Systematic Revision of Symbiodiniaceae Highlights the Antiquity and Diversity of Coral Endosymbionts, Current Biology)."*

We have revised the sentence to read: "In general, reef corals are highly adapted to oligotrophic waters with micro-algae symbionts to allow an efficient use of essential nutrients and to outcompete other fast-growing biota on a reef whose growth is inhibited by the undersupply of nutrients (Barrot and Rohwer, 2012; LaJeunesse et al., 2018; Vermeij et al., 2010)."

3.14. *"Lines 66-68: These sentences could be a bit stronger to introduce this important topic in your introduction."*

We have revised the sentence to read: "Understanding how coral calcification responds to rapid changes in seawater nutrient conditions and $\Omega_{sw}$ is critical for more accurate predictions on the persistence of reef habitats under the influence of global change."

3.15. *"Lines 70-72: This statement would benefit from a clear connection of how calcification responses from upwelling locations can be applied to systems without upwelling."*

We have revised the sentence to read: "This allows these regions to serve as natural laboratories to investigate the calcification response of reef corals to these multiple environmental stressors that are likely to affect global coral reefs in the near future (Camp et al., 2018; Wizemann et al., 2018)."

3.16. *"Lines 84-125: This section should be moved into the methods as a section describing the sites. A condensed version of this could be included in the previous paragraph to describe the sites if that is desired."*

We have moved the chapter "Arabian Sea climate and oceanography" to the beginning of the methodology section in the revised version of the manuscript.

**Methods**

3.17. *"Lines 128-134: It sounds like these colonies were not collected in situ, rather collected as dead skeletons from the beach. What about differences in local conditions? If these corals were washed up on shore you don't really know what depth or location they came from? Also what years? How do you know when they washed up on the beach?"*

We have streamlined this text section in the revised manuscript to clarify that the specimens presented are collected as dead coral material from a storm deposit associated with cyclone Gonu in 2007 (Fritz et al., 2010) (line 120-122).

The original growth position of the specimens was 1-4 metres below sea surface, as coral growth at Masirah Island is limited to these very shallow waters (Glynn, 1993) (see line 116-117). Using colonies from shallow waters for estimating mean coral calcification for a site is in line with other growth studies in the literature, considering colonies grown in water depths of 1-6 metres (Lough and Barnes, 2000; Manzello et al., 2014; Mollica et al., 2018). Depth related variation in coral growth is generally not expected in this near surface zone (Schlager, 1992).

It should also be mentioned here that substantial discussion in the revised manuscript focusses on intra-annual variability of calcification with regard to seasonal environmental changes (line 321-362). Location factors are hereby to be neglected, as they remain uniform within a single specimen.

The detailed information about the calendar years represented in our records is marginal, as we establish generalized annual calcification records with monthly resolution from several years of the total record lengths (Multi-year monthly means). These generalised records represent the mean monthly calcification of *Porites* corals at Masirah Island during the more recent current era.

3.18.  *"Lines 216-219: Unclear what these numbers and acronyms represent. Please rephrase in a clearer way."*

We have revised the sentence to read: "A detailed chronological frame for the Li/Mg records was established with the aid of the generalized annual record of remote sensing SST data (JPL MUR, daily averaged 2003-2018) (Fig.2). Dates of seasonal SST extremes as well as dates of inflection points between consecutive seasons were assigned to corresponding data points of the Li/Mg records (see supplementary material, Fig.S2)." The newly appended supplementary Fig.S2 provides a detailed illustration on how the age model was developed.

3.19.  *"Note on the standard corals: How do we know that the other corals are effective standards for comparison? Who is to say that they were not influenced by SST or OA? Or other factors?"*

We did not exclude the influence of environmental factors other than SST on the calcification of the Indo-Pacific reference corals (GBR and Hawaii) used for comparison in the preprint version of our manuscript at any point. Based on the highly convincing correlations with SST (bulk density: $r^2 = 0.49$, $p < 0.0001$ ; extension rate: $r^2 = 0.9$, $p < 0.0001$), we stated that calcification of these corals is "largely controlled" by water temperature (Lough and Barnes, 2000). However, we agree that our approach of applying the regressions from the Indo-Pacific reference dataset to predict calcification performance of *Porites* from the Arabian Sea possibly overstresses the global validity of the causal relationship between SST and calcification. For this reason, we have omitted the use of "standard corals" in the revised manuscript and instead provide a qualitative comparison of our data to an extended dataset of *Porites* calcification reported in the literature from other reefs in the Atlantic and Pacific (Fig.9).

**Results**

3.20.  *"Lines 248-249: Do these n refer to the number of transects for the measures?"*

These n referred to the number of years considered in the calculation of the mean annual and sub-annual (seasonally, monthly) calcification of individual specimens. In the revised manuscript, these numbers are presented in table form in order to provide a clear structure (Table 2).

**Discussion**

3.21.  *"A lot of the current discussion is results. These should be moved to the results section and then the discussion can include more incorporation of implications/meaning of these results."*

Thank you very much for this advice. We have moved several parts of the discussion (e.g., an updated version of Fig.9) to the results.

3.22.  *"Lines 285-288: But don't you expect biological variability?"*

This quantitative comparison of our data with data of *Porites* from the Indo-Pacific no longer appears in the revised version of our manuscript.

3.23. *"Lines 367-369: Split into two different sentences for easier reading."*

We have revised the section to read: "This finding implies that there is no intensified upregulation of internal $\Omega_{cf}$ relative to $\Omega_{sw}$ during the non-upwelling seasons (McCulloch et al., 2017; DeCarlo et al., 2018; D'Olivo and McCulloch, 2017). As an explanation, we propose that internal upregulation processes of corals affected by seasonal upwelling are not capable to adapt completely to ocean chemistry change on a quarterly scale. As a consequence, a relatively low $\Omega_{cf}$ is maintained year-round so as to avoid high gradients to the external $\Omega_{sw}$ during southwest monsoonal upwelling." (line 391-395)

3.24. *"Lines 414-433: This section is lacking incorporation of the current literature and needs some more grounding in terms of what is known and previous work."*

This section underwent substantial reorganisation and is no longer available.

**Figures**

3.25. *"Figure 1: This is a really helpful figure to demonstrate this reef system, collection site, and the currents/upwelling, however, this figure could be made a bit clearer with a few updates as suggested here. There is a lot going on with colours so I recommend making your land either white or grey to make the focus of the map more on the reef locations. I also recommend selecting a different colour to represent the coral reef provinces with better contrast against the red and blue."*

We have turned the colour of the land surface greyish to put more focus on the reef locations. The yellow colour of the coral reef provinces contrasts most strongly with the adjacent red and blue colours, which is why we consider it to be exceptionally suitable.

3.26. *"Figure 2/3: Please define NEM, SIM, SWM, and AIM in your figure captions."*

We have defined the abbreviations in the figure captions.

3.27. *"Figure 3: please include what years were assessed in to calculate these monthly values."*

The number of years included in calculating the monthly values (multi-year monthly means) are shown in Table 2 of the revised manuscript.

3.28. *"Figure 5: again, please include the years assessed in the monthly values."*

The number of years included in calculating the monthly values (multi-year monthly means) are shown in Table 2 of the revised manuscript.

**3. References**

Barrot, K.T., Rohwer, F.L.: Unseen players shape benthic competition on coral reefs. Trends Microbiol., 20, 621-628, http://doi.org/10.1016/j.tim.2012.08.004, 2012.

Camp, E.F., Schoepf, V., Mumby, P.J., Hardtke, L.A., Rodolfo-Metalpa, R., Smith, D.J., Suggett, D.J.: The future of coral reefs subject to rapid climate change: Lessons from natural extreme environments. Front. Mar. Sci., 5, 1–21, https://doi.org/10.3389/fmars.2018.00004, 2018.

Cornwall, C.E., Comeau, S., Kornder, N.A., Perry, C.T., van Hooidonk, R., DeCarlo, T.M., Pratchett, M.S., Anderson, K.D., Browne, N., Carpenter, R., Diaz-Pulido, G., D'Olivo, J.P., Doo, S.S., Figueiredo, J., Fortunato, S.A.V., Kennedy, E., Lantz, C.A., McCulloch, M.T., González-Rivero, M., Schoepf, V., Smithers, S.G., Lowe, R.J.: Global declines in coral reef calcium carbonate production under ocean acidification and warming. Proc. Natl. Acad. Sci. U. S. A., 118, 1-10, https://doi.org/10.1073/pnas.2015265118, 2021.

Courtney, T.A., Lebrato, M., Bates, N.R., Collins, A., De Putron, S.J., Garley, R., Johnson, R., Molinero, J.C., Noyes, T.J., Sabine, C.L., Andersson, A.J.: Environmental controls on modern scleractinian coral and reef-scale calcification. Sci. Adv. 3, 1-9, https://doi.org/10.1126/sciadv.1701356, 2017.

Garcia, H. E., Weathers, K., Paver, C. R., Smolyar, I., Boyer, T. P., Locarnini, R. A., Zweng, M. M., Mishonov, A. V., Baranova, O. K., Reagan, J. R.: World Ocean Atlas 2018, 4: Dissolved Inorganic Nutrients (phosphate, nitrate, silicate), 2019.

Glynn, P.W.: Monsoonal upwelling and episodic *Acanthaster* predation as probable controls of coral reef distribution and community structure in Oman, Indian Ocean. Atoll Res. Bull., 379, 1–66. https://doi.org/10.5479/si.00775630.379.1, 1993.

Guan, Y., Hohn, S., Wild, C., Merico, A.: Vulnerability of global coral reef habitat suitability to ocean warming, acidification and eutrophication. Glob. Chang. Biol., 26, 5646–5660, https://doi.org/10.1111/gcb.15293, 2020.

Hall, E.R., Muller, E.M., Goulet, T., Bellworthy, J., Ritchie, K.B., Fine, M.: Eutrophication may compromise the resilience of the Red Sea coral Stylophora pistillata to global change. Mar. Pollut. Bull., 131, 701–711, https://doi.org/10.1016/j.marpolbul.2018.04.067, 2018.

LaJeunesse, T.C., Parkinson, J.E., Gabrielson, P.W., Jeong, H.J., Reimer, J.D., Voolstra, C.R., Santos, S.R.: Systematic Revision of Symbiodiniaceae Highlights the Antiquity and Diversity of Coral Endosymbionts. Curr. Biol., 28, 2570-2580, https://doi.org/10.1016/j.cub.2018.07.008, 2018.

Lough, J.M., Barnes, D.J.: Environmental controls on growth of the massive coral *Porites*. J. Exp. Mar. Bio. Ecol., 245, 225-243, http://doi.org/10.1016/s0022-0981(99)00168-9, 2000.

Montaggioni, L.F., Le Cornec, F., Corrège, T., Cabioch, G.: Coral barium/calcium record of mid-Holocene upwelling activity in New Caledonia, South-West Pacific. Palaeogeogr., Palaeoclimatol., Palaeoecol., 273, 436-455, https://doi.org/10.1016/j.palaeo.2005.12.018, 2005.

Ross, C.L., DeCarlo, T.M., McCulloch, M.T.: Environmental and physiochemical controls on coral calcification along a latitudinal temperature gradient in Western Australia. Glob. Chang. Biol., 25, 431–447, https://doi.org/10.1111/gcb.14488, 2019.

Schlager, W.: Sedimentology and sequence stratigraphy of reefs and carbonate platforms: A short course. Am. AAPG Bulletin, 34, https://doi.org/10.1306/CE34551, 1992.

Tomascik, T., Sander, F.: Effects of eutrophication on reef-building corals. Mar. Biol., 87, 143–155, https://doi.org/10.1016/0198-0254(87)90298-6, 1985.

Tomascik, T.: Growth rates of two morphotypes of *Montastrea annularis* along a eutrophication gradient, Barbados, W.I. Mar. Pollut. Bull., 21, 376–381, https://doi.org/10.1016/0025-326X(90)90645-O, 1990.

Tudhope, A.W., Lea, D.W., Shimmield, G.B., Chilcott, C.P., Head, S.: Monsoon climate and Arabian Sea coastal upwelling recorded in massive corals from southern Oman. Palaios, 11, 347-361, https://doi.org/10.2307/3515245, 1996.

Vermeij, M.J.A., van Moorselaar, I., Engelhard, S., Hörnlein, C., Vonk, S.M., Visser, P.M.: The effect of nutrient enrichment and herbivore abundance on the ability of turf algae to overgrow coral in the Caribbean. PLoS One, 5, 1-8, https://doi.org/10.1371/journal.pone.0014312, 2010.

Wizemann, A., Nandini, S.D., Stuhldreier, I., Sánchez-Noguera, C., Wisshak, M., Westphal, H., Rixen, T., Wild, C., Reymond, C.E.: Rapid bioerosion in a tropical upwelling coral reef. PLoS One, 13, 1–22, https://doi.org/10.1371/journal.pone.0202887, 2018.

---

## Editor Decision (ED1)

Overall, there are some interesting results here. I think your findings on P/N ratio and extension rate are the most interesting in this paper, however, it seems a bit overshadowed by your discussion on external and internal saturation state. Unfortunately, you just don't have the data to support your long discussion on that topic, although it makes a nice hypothesis. You need some sort of saturation state data to back up your claims about low omega reducing calcification. If you can't find anything in this region, I suggest you remove the discussions focused on this and rewrite as one paragraph introducing your hypothesis. There is plenty of interesting data in the relationship with P/N that I don't think the long discussion about saturation state is completely necessary. Check the following sites for carbon chemistry data:

https://www.socat.info/
https://www.ncei.noaa.gov/access/ocean-carbon-acidification-data-system/
https://www.bco-dmo.org/project/2099

Check grammar and word usage carefully throughout, there are a lot of strange word choices and grammatical errors. Some specific examples are below, but please make sure to read your manuscript carefully and look for other errors.

L31: I wouldn't use 'coralline' here as it could be confusing with crustose coralline algae

L34: Replace 'on' with 'to', remove 'contemporarily'

L54: I think 'savage' should be 'sewage'

L55: Replace 'versatile' with 'diverse'

L63: This is a bit confusing as written, please rewrite to make clear that eutrophication along the off-inshore gradient is inhibiting calcification where there are more nutrients.

L61: Use another word than 'versatile', e.g. diverse, different, various

L68: 'Heralded' is an awkward word choice here

L74: Remove 'solely'

L76: Start a new paragraph with 'In the here presented study…' Also, reword that phrase to 'In this study'

L136: Is 'tournament' the correct word here?

L180: Change 'accordance' to 'agreement'

L269: 'Curse' should be 'curve'

L273: Why is 'forereef' in the reference?

L277: 'A generally enhanced…' reword this sentence to make it more clear what you are trying to say

Fig. 6 legend: Should coral growth temperatures just be sea surface temperatures?

L286: Font or text size change is apparent in the pdf

L309: 'Nonetheless, published…' this sentence needs to be reworded for clarity

L311: Remove 'however' after 'in contrast'

L314: '…were well related…' awkward phrasing

L314: 'quality of correlation' what correlation specifically?

L331: 'brought up no unambiguous' awkward phrasing

L364: Replace 'rather' with 'likely'

L365: I would refrain from citing  personal communication about seawater chemistry data, you could rephrase as a hypothesis

L426: 'The here presented data' change to 'Our data'

L439: Remove 'too'

Section 3.2: The use of abbreviations makes reading comprehension a bit difficult at this point. I would suggest that you try to avoid using so many abbreviations and write some of these out. For example I couldn't remember offhand what SIM, AIM, SWM, and NEM while reading this section. Try to only use easy to understand and general abbreviations (e.g., SST).

---

## Author Response (AR2)

Point-by-point description of changes made to the revised manuscript

**"Calcification response of reef corals to seasonal upwelling in the northern Arabian Sea (Masirah Island, Oman)"**
Philipp M. Spreter et al.

*Biogeosciences*, doi.org/10.5194/bg-2021-213, 2021

We would like to thank the editor and reviewer #3 for commenting on our manuscript. Their supporting comments and ideas have substantially improved the scientific significance of our work. In line with the reviewers' and editors' comments, minor changes have been made to our revised manuscript. Generally, all suggestions for the improvement of language and grammar were implemented. Detailed responses to the specific comments made on our manuscript are given in a point-by-point manner in the following.

**1. Editor comments:**

*1.1*      *"Lines 62-64: This is a bit confusing as written, please rewrite to make clear that eutrophication along the off-inshore gradient is inhibiting calcification where there are more nutrients."*

We have revised the sentence to clarify that increasing eutrophy causes a "stretching modulation of the skeleton". This modulation in the patterns of calcification can result either in reduced, constant (Carricart-Ganivet and Merino, 2001; Carricart-Ganivet, 2004) or enhanced (D'Olivo et al., 2013; Manzello et al., 2015) rate of calcification.

The revised the sentence is to read: "In general, increasing eutrophy is considered to cause reef corals to sacrifice skeletal density for increased extension rate ("stretching modulation of skeletal growth"), which can either lead to enhanced, constant or reduced rate of calcification (Carricart-Ganivet and Merino, 2001; Carricart-Ganivet, 2004; D'Olivo et al., 2013; Manzello et al., 2015).

*1.2*      *"Line 136: Is "tournament" the correct word here?"*

The word "tournament" was replaced by the word "rotation".

*1.3*      *"Line 327: Reduce the use of acronyms, these get very confusing."*

Acronyms for the individual seasons (i.e. SWM, AIM, NEM, SIM) are now written out as whole words for better readability.

*1.4*      "Line 365: I would refrain from citing personal communication about seawater chemistry data, you could rephrase as a hypothesis."

In this sentence, we refer to an unpublished dataset, which provides sub-annually resolved $\Omega_{cf}$ (based on combined $\delta^{11}B$ and B/Ca analyses) derived from a *Porites* collected within the upwelling zone of Panama (Saboga). Annual means in $\Omega_{cf}$ of the same coral are published in Mollica et al., 2018 (see Fig.1; Saboga). The unpublished sub-annual data demonstrate $\Omega_{cf}$ to be relatively low all year round, even during the non-upwelling season. These data are an important contribution to our hypothesis suggesting the year-round low skeletal density of corals from the Arabian Sea upwelling zone to be the result of a constantly low $\Omega_{cf}$. After consultation with N. Mollica, we have provided the data on the sub-annually resolved $\Omega_{cf}$ as a supplement on which we refer to within the text (see supplementary material, Fig. S3).

*1.5*    *"Line 367-370: This sentence is very confusing. Rewrite, and maybe break into two sentences."*

We have revised the sentence to read: "A similar finding is reported from two sites located within the Galapagos upwelling zone (n = 7-8 cores per site) (Manzello et al., 2014). Poor replication of *Porites* calcification data from the upwelling areas of Panama and the South China Sea (n = 1, respectively) does not enable a proper comparison (Mollica et al., 2018).

*1.6*    *"Line 378-380: Wording is a bit strange here, rewrite this portion."*

We have revised the sentence to read: "Hence, a stimulating effect of nutrients on extension rate during the non-upwelling seasons is possible, since moderate nutrient concentrations with high $PO_4^{3-}$ to $NO_3^-$ ratio exist year-round in the Arabian Sea (Dunn et al., 2012; Kleypas et al., 1999; Koop et al., 2001)."

*1.7*    *"Line 386-387: Awkward wording."*

We have revised the sentence to read: "Accordingly, we hypothesise the year-round relatively low skeletal density of the Masirah corals to be also related to a constantly low $\Omega_{cf}$."

*1.8*    *"Line 414: Add a sentence after describing how future work may confirm this hypothesis."*

We added the sentence to the end of the conclusion section to read: "Further research should include combined analyses of $\delta^{11}B$ and B/Ca ratios in order to confirm the hypothesis of a year-round relatively low $\Omega_{cf}$ in reef corals from sites affected by seasonal upwelling."

**2   Reviewer comments (RC #3):**

**General comments:**

*2.1*    *"Unfortunately, there is low sample size for the sub-annual measurements. This results in high variability of the calcification data. Even though the authors have included additional samples showing similar annual bulk density, extension rate, and calcification rate, without the sub annual data it does not help to clear up many uncertainties.."*

Intra-reef variability in the patterns of calcification between individual coral colonies is largely related to differences in locational conditions (e.g. depth, illumination, nutrients, etc.). These locational differences are negligible when comparing internal, sub annual growth patterns of one single coral. Sub-annual variability in calcification performance within a single specimen reflects seasonal changes in environmental conditions at a particular site.

Consistently, all three sub-annually resolved specimens from Masirah Island reveal lower extension (calcification) rates during the upwelling season compared to the winter months (northeast monsoon) (with approximately similar SSTs) (see table 3). We have no doubt on this being a seasonal environmental signal rather than being an artefact of intra-reef variability. Therefore, we have no reason for concern about "many uncertainties".

*2.2*    *"One of the major flaws in its current form is that most conclusions are derived from data that is not included or were not measured i.e. calcification fluid carbonate chemistry, seawater conditions for turbidity, light, or complete carbonate chemistry.."*

We agree to reviewer #3 that assumptions about $\Omega_{cf}$ in the discussion section and conclusion section of our manuscript cannot be backed up by analytical data. For this reason, assumptions concerning $\Omega_{cf}$ will appear as hypotheses in the revised version of our manuscript (see changes made to line 387 and following). However, we strongly feel that there is no necessity to measure turbidity, light levels, nutrient concentrations or seawater carbonate chemistry since information on these parameters are available in published literature (please see Quinn and Johnson, 1996; Takahashi et al., 2014; Garcia et al., 2019).

**Detailed comments:**

2.3     *"Line 54-55: Incomplete sentence"*

We have revised the sentence to read: "Eutrophication can have both beneficial as well as detrimental effects on coral growth, depending on the kind of available nutrients and their concentration (Tomascik and Sander, 1985; Tomascik, 1990)."

2.4     *"Line 55-56: Consider rewording this sentence"*

We have revised this section to read: "In general, reef corals are highly adapted to oligotrophic waters with micro-algae symbionts to allow an efficient use of essential nutrients (Muscatine and Porter, 1977). This enables outcompeting other fast-growing biota on a reef whose growth is inhibited by the undersupply of nutrients (Vermeij et al., 2010; Barott and Rohwer, 2012)."

2.5     *"Line 98-99: But still supersaturated with respect to aragonite during the upwelling season?"*

Indeed, the seawater remains supersaturated with respect to aragonite even during the upwelling season. However, saturation drops below critical values assumed to be required for coral growth (Kleypas et al., 1999).

2.6     *"Line 120: I feel there is a lack of description from where/how the cores were collected since position on the reef can be a big factor for many of the points measured here."*

We strongly feel that the origin and collection of the specimens is adequately addressed in section 2.2.

2.7     *"Lines 139-140: How were these 3 corals selected out of the 6?"*

We have revised this section to read: "Corals 5.10, 5.13 and 5.21 were selected for Li/Mg and Ba/Ca geochemical analysis based on optimal orientation and traceability of the corallites adjacent to the density measurement transects."

2.8     *"Line 327-328: "All specimens show lower SWM calcification rates compared to NEM." is this accurate? I would argue that coral 5.13 is identical in SWM and NEM and coral 5.21 shows too much variation to say its different."*

According to seasonal means and standard deviations shown in Table 3, both, coral 5.13 as well as coral 5.21 have lower calcification rates during the southwest monsoon than during the northeast monsoon.

2.9     *"Lines 369: What identical results?"*

We have revised the sentence to read: "A similar finding is reported from two sites located within the Galapagos upwelling zone (n = 7-8 cores per site) (Manzello et al., 2014). Poor replication of *Porites* calcification data from the upwelling areas of Panama and the South China Sea (n = 1, respectively) does not enable a proper comparison (Mollica et al., 2018).

2.10     *"Lines 386-387: But skeletal density shows an increase to SWM upwelling? And based on your summary of how skeletal density might be related to $\Omega_{cf}$ – this would mean $\Omega_{cf}$ is increasing in response to an external decrease, potentially to offset negative effects?"*

All three sub-annually resolved coral specimens show a high-density band during the upwelling season. No evidence exists, however, that the density bands are driven by up-regulation of the $\Omega_{cf}$ during the upwelling season. Instead, figure 7 shows the high skeletal density during southwest monsoon to be a function of decreasing extension rate ($r^2 = 0.88$, p = 0.0002) (see also line 270 – 275). As an explanation, we refer to the work of DeCarlo and Cohen (2017), showing that with decreasing extension rate, more carbonate is precipitated in less space, which leads to higher skeletal density (line 351 – 357).

2.11 *"Line 388-389: I see what you are saying but this is not a strong argument. Highly suggest to remove an Li/Mg ratio offset as evidence that pH$_{cf}$ is constant"*

We have revised the sentence to ensure the reader is aware that the high Li/Mg ratios support, but do not confirm the hypothesis of a low $\Omega_{cf}$. The revised sentence reads: "This hypothesis is further supported by the Li/Mg ratios of the Masirah corals, which are offset to higher values than those expected from the literature".

2.12 *"Lines 393-394: the statement "not capable to adapt completely to ocean chemistry change on a quarterly scale" seems to not have much basis on the limited data you provide, particularly since there is no internal saturation data. It would rather seem that external saturation state is not a good determination for calcification in general since you show that overall calcification is indistinguishable from calcification data in non-upwelling zones elsewhere."*

The sentence annotated provides an approach to explain the absence of an immediate response in the intra-annual density variability to temporary low $\Omega_{sw}$ during the seasonal upwelling. Although no immediate response to the upwelling is observed, the mean annual skeletal density is lower than in corals from sites unaffected by upwelling. According to actual literature, we therefore propose $\Omega_{cf}$ of the Masirah corals to be kept relatively constant by modification, but to be less upregulated than in corals from regions without upwelling (McCulloch et al., 2017; Mollica et al., 2018).

2.13 *"Line 395: I agree, $\Omega_{cf}$ is probably held constant throughout the year, however it is more likely another constant factor such as high year round nutrients, which at relative high levels promotes extension/limits density (as you cite from Manzello et al 2014) and then is reversed under upwelling either from turbidity as you suggest or extreme high nutrients or both."*

As shown in Figure 9c, the skeletal density of *Porites* from upwelling areas is consistently lower compared to those unaffected by upwelling. In contrast, extension rates at the same sites are highly variable, with relatively slow growth in Panama and fast growth at a site off Galapagos (Fig. 9b). We therefore consider both the seasonal dynamics of seawater nutrients (as discussed in lines 339-349; 361-362; 385-390) and a constantly relatively low $\Omega_{cf}$ to determine the density patterns in corals from upwelling areas.

Further evidence for low skeletal density in corals from upwelling areas being linked to a relatively low $\Omega_{cf}$ was also demonstrated by combined analyses of $\delta^{11}$B and B/Ca for specimens from Saboga and the South China Sea (Mollica et al., 2018).

**3. References**

[revised manuscript text omitted]

---

## Author Response (AR3)

Response to the editor regarding corrections to grammatical issues in the manuscript

**"Calcification response of reef corals to seasonal upwelling in the northern Arabian Sea (Masirah Island, Oman)"**

Philipp M. Spreter et al.

*Biogeosciences*, doi.org/10.5194/bg-2021-213, 2021

Dear Dr. Cyronak,

we greatly appreciate the opportunity to correct some grammatical issues within our manuscript prior to final publication. All suggestions for the improvement of language and grammar were implemented and revisions were made within two sections of the text, which are outlined in the following.

**1. Editor suggestions:**

1.1      *"L26: Reword last sentence as it is a bit confusing."*

We have revised the sentence to read:

"Variable responses of reef coral extension to nutrients, which either exacerbate or compensate negative effects of diminished skeletal density associated with ocean acidification, may therefore be critical to the maintenance of adequate carbonate accumulation rates in coral reefs under global change."

1.2      *"L414 (last paragraph): I would put the last sentence that begins on L420 after the first sentence in the last paragraph (starting on L414 with 'These results…'). I think the other two are stronger sentences to end with."*

According to the editor's suggestion, we have revised this paragraph to read:

"These results suggest that temporarily reduced $\Omega_{sw}$ (seasonal upwelling) has no instantaneous impact on sub-annual variability in skeletal density but could cause a permanent adaptation towards year-round unexpected low skeletal density. Further research should include combined analyses of $\delta^{11}B$ and B/Ca ratios in order to confirm the hypothesis of the permanently low skeletal density in reef corals from sites affected by seasonal upwelling is controlled by a year-round comparatively low $\Omega_{cf}$. Unless the low skeletal density is compensated through high extension rate, this will yield detrimental effects on the net carbonate accumulation in coral reefs. Furthermore, this study highlights variable effects of nutrients on extension rate, with negative effects at excessively high nutrient levels (i.e., upwelling season) and stimulatory effects at moderate nutrient levels (i.e., non-upwelling season)."